# Comparative epidemic expansion of SARS-CoV-2 variants Delta and Omicron in the Brazilian State of Amazonas

Ighor Arantes[1,2,16], Gonzalo Bello [1,16] ✉, Valdinete Nascimento[3], Victor Souza[3], Arlesson da Silva[3], Dejanane Silva[3], Fernanda Nascimento[3], Matilde Mejía[3], Maria Júlia Brandão[3], Luciana Gonçalves[3,4], George Silva[3,5], Cristiano Fernandes da Costa[4,15], Ligia Abdalla[6], João Hugo Santos[7], Tatyana Costa Amorim Ramos[4], Chayada Piantham[8], Kimihito Ito[9], Marilda Mendonça Siqueira[2], Paola Cristina Resende [2], Gabriel Luz Wallau [10,11], Edson Delatorre[12], Tiago Gräf[13] & Felipe Gomes Naveca [3,14] ✉

The SARS-CoV-2 variants of concern (VOCs) Delta and Omicron spread globally during mid and late 2021, respectively. In this study, we compare the dissemination dynamics of these VOCs in the Amazonas state, one of Brazil's most heavily affected regions. We sequenced the virus genome from 4128 patients collected in Amazonas between July 1st, 2021, and January 31st, 2022, and investigated the viral dynamics using a phylodynamic approach. The VOCs Delta and Omicron BA.1 displayed similar patterns of phylogeographic spread but different epidemic dynamics. The replacement of Gamma by Delta was gradual and occurred without an upsurge of COVID-19 cases, while the rise of Omicron BA.1 was extremely fast and fueled a sharp increase in cases. Thus, the dissemination dynamics and population-level impact of new SARS-CoV-2 variants introduced in the Amazonian population after mid-2021, a setting with high levels of acquired immunity, greatly vary according to their viral phenotype.

The SARS-CoV-2 variants of concern (VOCs) Delta and Omicron have spread worldwide and reached dominance in mid- and late 2021, respectively[1,2]. The first Omicron lineage to spread worldwide was BA.1, followed by BA.2 and BA.4/5[3,4]. While Delta displayed large heterogeneity in dissemination dynamics between locations, Omicron BA.1 spread displayed much less variability globally[5,6].

A previous study predicted that the population-level impact of new SARS-CoV-2 variants in immunized populations might vary

[1]Laboratório de AIDS e Imunologia Molecular, Instituto Oswaldo Cruz, Fiocruz, Rio de Janeiro, Brazil. [2]Laboratório de Vírus Respiratórios, Exantemáticos, Enterovírus e Emergências Virais, Instituto Oswaldo Cruz, Fiocruz, Rio de Janeiro, Brazil. [3]Laboratório de Ecologia de Doenças Transmissíveis na Amazônia, Instituto Leônidas e Maria Deane, Fiocruz, Manaus, Brazil. [4]Fundação de Vigilância em Saúde do Amazonas - Dra Rosemary Costa Pinto, Manaus, Brazil. [5]Fundação Centro de Controle de Oncologia do Estado do Amazonas, Manaus, Brazil. [6]Universidade do Estado do Amazonas, Manaus, Brazil. [7]Hospital Adventista de Manaus, Manaus, Brazil. [8]Graduate School of Infectious Diseases, Hokkaido University, Hokkaido, Japan. [9]International Institute for Zoonosis Control, Hokkaido University, Hokkaido, Japan. [10]Instituto Aggeu Magalhães, Fundação Oswaldo Cruz, Recife, Brazil. [11]Department of Arbovirology, Bernhard Nocht Institute for Tropical Medicine, Hamburg, Germany. [12]Departamento de Biologia, Centro de Ciências Exatas, Naturais e da Saúde, Universidade Federal do Espírito Santo, Alegre, Brazil. [13]Laboratório de Virologia Molecular, Instituto Carlos Chagas, Fiocruz, Curitiba, Brazil. [14]Laboratório de Arbovírus e Vírus Hemorrágicos, Instituto Oswaldo Cruz, Fiocruz, Rio de Janeiro, Brazil. [15]Present address: Conselho de Secretários Municipais de Saúde do Amazonas COSEMS – AM, Manaus, Brazil. [16]These authors contributed equally: Ighor Arantes, Gonzalo Bello. ✉e-mail: gbellobr@gmail.com; felipe.naveca@fiocruz.br

according to their phenotypes[7]. Viral variants that mainly display enhanced intrinsic transmissibility will lead to small numbers of reinfections/breakthrough infections, which may limit the epidemic's size. In contrast, viral variants that combine increased transmissibility and immune escape features will produce significant numbers of reinfections/breakthrough infections, and the epidemic size is expected to be markedly increased without nonpharmaceutical interventions (NPIs). The dissemination dynamics of SARS-CoV-2 variants may be also shaped by the immune population landscape and populations with high levels of "hybrid immunity", produced by a combination of natural infection and vaccination, are expected to have the best protection[8,9]. Hybrid immunity improves the potency, breadth, and durability of serum-neutralizing activity against SARS-CoV-2 variants, including Delta and Omicron, relative to either infection or vaccination alone[10–15].

This study aimed to compare the population-level impact and dissemination dynamics of SARS-CoV-2 Delta and Omicron BA.1 variants spreading in Amazonas state, a region with high levels of hybrid immunity. Delta and Omicron BA.1 variant's relative growth has been documented in countries with high levels of either natural (like South Africa) or vaccine (like the United Kingdom [UK]) acquired immunity[3,16–18], but their epidemic dynamics in locations with high levels of hybrid immunity remained poorly described. The Brazilian state of Amazonas was one of South America's most heavily affected regions[19,20] and it was estimated that ≥70% of the population was already infected by March 2021[21,22]. Moreover, vaccination roll-out started in the Amazonas in February 2021 and 50% and 70% of the population received at least one vaccine dose by mid- and late 2021, respectively[23]. Thus, when Delta and Omicron BA.1 variants were first detected in the state in July and December 2021, respectively, they encountered a population with high levels of preexisting hybrid immunity.

## Results

### Spatiotemporal distribution of SARS-CoV-2 VOCs in Amazonas
In this study, we generated 4128 SARS-CoV-2 high-quality, whole-genome sequences from individuals living in all 11 Amazonas state regions between July 1, 2021, and January 31, 2022, representing 3.2% of all laboratory-confirmed SARS-CoV-2 cases ($n = 127,787$) in the state in that period (Fig. 1a). In line with the geographic location of the SARS-CoV-2 positive cases, most of the genomes were from the Manaus metropolitan area (79%), which comprises the capital city and other surrounding municipalities, followed by the regions of Parintins (6%), Manacapuru (6%), Tefé (3%), Itacoatiara (1%), and Tabatinga (2%) (Fig. 1b, c). Out of the 4128 SARS-CoV-2 genomes generated, 1024 (25%) were classified as Gamma (P.1/P.1.*), 965 (23%) as Delta (B.1.617.2/AY.*), 2135 (52%) as Omicron (B.1.1.529/BA.1.*) and four (<1%) as Mu (B.1.621). The temporal pattern revealed a gradual replacement of VOC Gamma by Delta followed by a rapid substitution of Delta by Omicron (Fig. 1d). Gamma was the most prevalent (>50%) variant in the Amazonas state until mid-September and was detected for the last time on November 16, 2021. Delta was first detected in the state on July 21, 2021 and took ~70 days to become the most prevalent (>50%) variant and 100–110 days to become the dominant (>90%) one. Omicron BA.1 was first detected in Amazonas on December 21, 2021 and rapidly increased in frequency, taking only ~2 weeks to become the dominant (>90%) variant. The time interval from the first detection to dominance for Omicron BA.1 (~10–20 days) was much shorter than for Gamma (~40–50 days) and Delta (~100–110 days) variants in Amazonas (Fig. 1e). Delta and Omicron BA.1 were first detected in Manaus city (the Capital) and only later in the countryside municipalities. Since the first detection, Delta took ~80–90 days and >120 days to become dominant in Manaus and outside Manaus, respectively; while Omicron BA.1 took <30 days to become the dominant circulating variant within and outside Manaus (Fig. 1f).

### Epidemic dynamics of SARS-CoV-2 VOCs Delta and Omicron
The SARS-CoV-2 epidemic in Amazonas has been characterized by three COVID-19 epidemic waves with exponential growth in cases during the first 24 months (Fig. 2a). The first exponential wave between April and July 2020 was driven by descendants of the B.1 lineage (B.1.195 and B.1.1.28), and the second one between December 2020 and March 2021 by the VOC Gamma. The months following the second wave were characterized by an endemic-like transmission pattern with a roughly constant incidence of ~100–500 SARS-CoV-2 cases per day. This period was characterized by the gradual replacements of Gamma (P.1) by Gamma descendants lineages (mainly P.1.4 and P.1.6) between April and July 2021 and the subsequent replacement of Gamma descendent lineages by Delta between August and December 2021. This endemic-like period was followed by the third exponential wave that was immediately preceded by the rapid rise of the Omicron BA.1 variant that fueled a sharp upsurge in the mean number of daily SARS-CoV-2 cases from ~90 to ~6500 during January 2022.

The Omicron BA.1 wave displayed three significant differences compared with previous waves in the Amazonas state. First, as a result of the rapid rise in infections, the incidence of new daily SARS-CoV-2 cases at the peak of the Omicron BA.1 wave (6493) was nearly four-fold and two-fold higher than at the peak of the B.1 (1696) and Gamma (2927) waves, respectively (Fig. 2a, d). Second, the overall duration of the Omicron BA.1 wave was much shorter than previous ones. The B.1 and Gamma epidemic waves displayed a mean time of ~60 days from the onset of case growth to their peaks and another ~60 days from their peaks to basal levels (<800 daily cases). The peak of the Omicron BA.1 wave occurred ~35 days after the first Omicron infection was identified and took another ~35 days to return to basal levels (Fig. 2a, d). Third, the number of deaths during the Omicron BA.1 wave was much lower than during previous waves (Fig. 2b, e), resulting in a substantial reduction in the case-fatality ratio (CFR), calculated as the ratio between the number of confirmed deaths and the number of confirmed cases 10 days earlier (Fig. 2c). The overall CFR estimated during the Omicron BA.1 wave (0.17) was much lower than that registered during B.1 (3.2) and Gamma (4.1) epidemic waves and was also lower than that observed during the endemic-like period of circulation of Gamma lineages P.1.4/P.1.6 and Delta (1.6–1.7).

### Estimating Delta and Omicron introductions into Amazonas
Most (99%) Delta and Omicron BA.1 genomes analyzed were grouped in five and nine Pango lineages, respectively (Table S1). Of the 965 Delta sequences here analyzed, 496 (52%) were classified as AY.99.2, 147 (15%) as AY.122, 107 (11%) as AY.43, 105 (11%) as AY.9.2, 57 (6%) as AY.101, and 53 (5%) were distributed among other 17 Delta lineages (B.1.617.2/AY.*). Of the 2135 Omicron BA.1 sequences identified, 1132 (53%) were classified as BA.1, 355 (17%) as BA.1.17.2, 204 (10%) as BA.1.1.15, 115 (5%) as BA.1.9, 82 (4%) as BA.1.14.1, 72 (3%) as BA.1.1, 51 (2%) as BA.1.20, 48 (2%) as BA.1.15, 37 (2%) as BA.1.14.2 and 39 (2%) were distributed among other eight Omicron lineages (BA.1.*). Lineages AY.99.2 and AY.101 were characterized as being the most prevalent Delta variants of Brazilian origin circulating in the country[24,25]; while lineages BA.1.9 and BA.1.14.* were recognized as BA.1 variants mostly composed by Brazilian sequences (https://cov-lineages.org/lineage_list.html).

To quantify the number of SARS-CoV-2 introductions from abroad into Amazonas, we analyzed sequences of the 14 most prevalent Delta and Omicron lineages combined with their closest worldwide relatives and subjected them to Bayesian phylogeographic analyses (Figs. S1–S7). Our analyses detected multiple introductions of Delta and Omicron BA.1 lineages into Amazonas that peaked in August–September 2021 and the first week of January 2022, respectively (Fig. 3a, b). The median estimated number of Omicron BA.1 ($n = 490$) introduction events into Amazonas was larger than Delta ($n = 120$) but greatly varied across lineages ranging from one (95% HPD

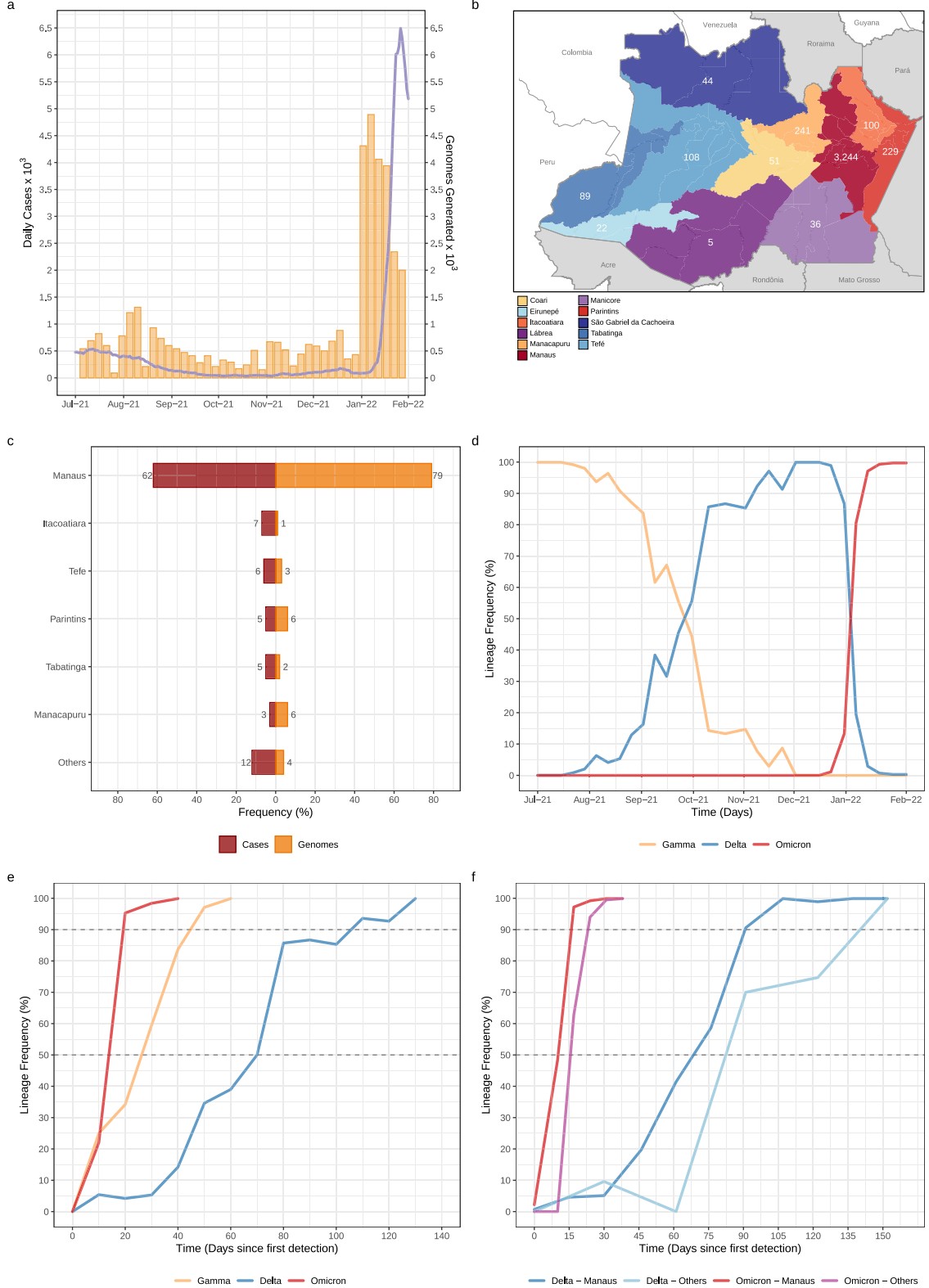

**Fig. 1 | Temporal and geographic distribution of SARS-CoV-2 cases and genomes in Amazonas state. a** Rolling average (7-day) of daily new SARS-CoV-2 confirmed cases in Amazonas (solid line) and the number of SARS-CoV-2 genomes sequenced in this study (yellow bars) between 1st July 2021 and 31st January 2022. **b** Map of Amazonas state showing regions covered by the SARS-CoV-2 genomes generated in this study. Numbers represent genomes generated for each region. **c** Number of SARS-CoV-2 confirmed cases and SARS-CoV-2 genomes from different Amazonian regions between 1st July 2021 and 31st January 2022. **d**–**e** Frequency of VOCs Gamma, Delta, and Omicron BA.1 among SARS-CoV-2 positive samples sequenced in Amazonas since 1st July 2021 (**d**) or since the first detection of each viral variant (**e**). **f** Frequency of VOCs Delta and Omicron BA.1 among SARS-CoV-2 positive samples sequenced within and outside Manaus since the first detection of each viral variant. Amazonas' map was generated with QGIS v.3.10.2 software (http://qgis.org) using public access data downloaded from the GADM v.3.6 database (https://gadm.org) and shapefiles obtained from the Brazilian Institute of Geography and Statistics (https://portaldemapas.ibge.gov.br/portal.php#homepage).

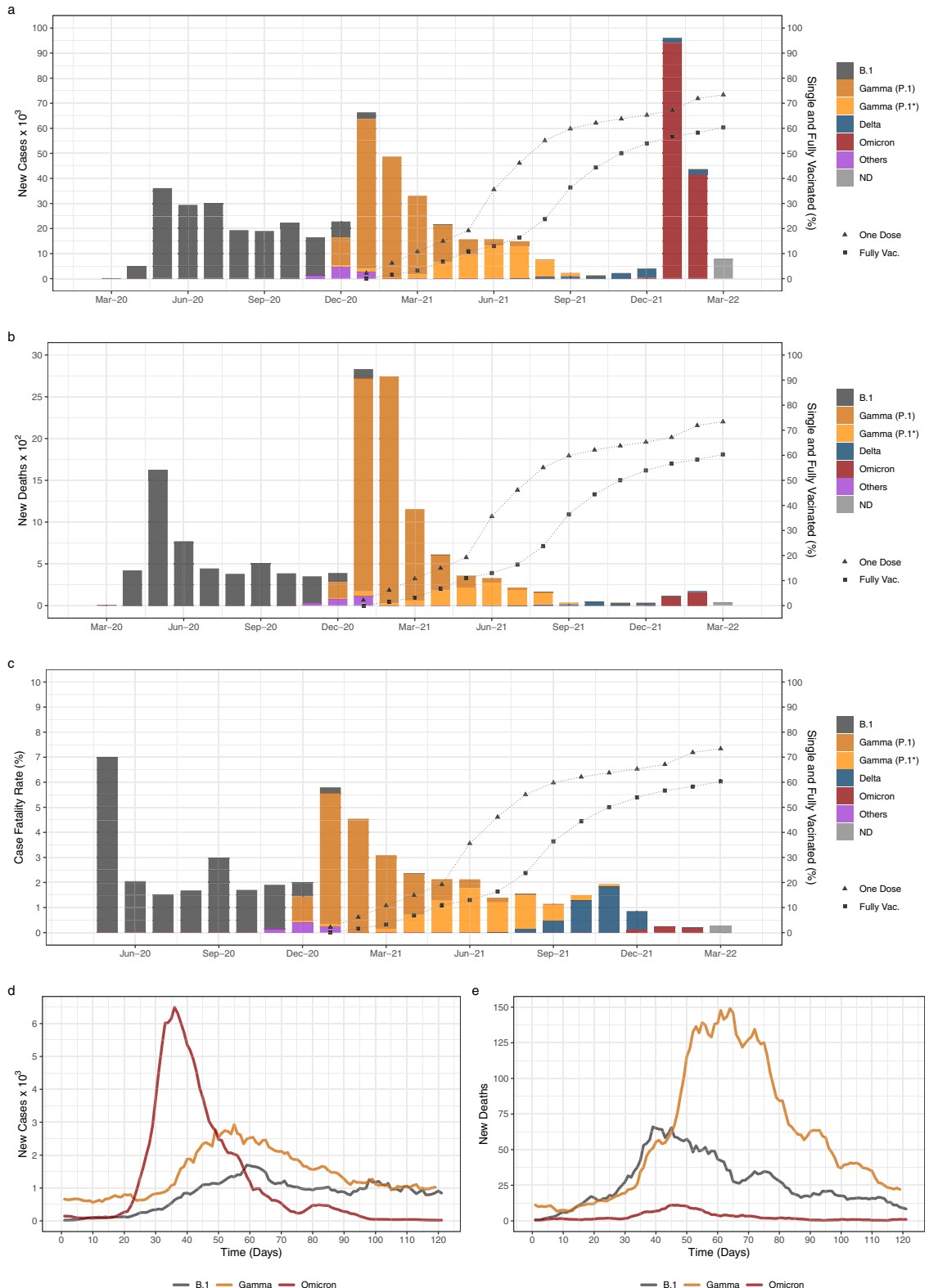

**Fig. 2 | Epidemic trajectories of SARS-CoV-2 variants circulating in Amazonas. a–c** Number of monthly confirmed SARS-CoV-2 cases (**a**), deaths (**b**), and case-fatality ratio (ratio between the number of confirmed deaths and confirmed cases)[80] (**c**) are represented together with the percentage of people single and fully vaccinated in Amazonas between March-2020 and March-2022. The color indicates the inferred relative prevalence of viral variants responsible for the infections and deaths as calculated from genomic surveillance data from the EpiCoV database at GISAID. **d, e** Number of confirmed new SARS-CoV-2 cases (**d**) and associated deaths (**e**) during the expansion of B.1, Gamma, and Omicron BA.1 variants in Amazonas since their first detection.

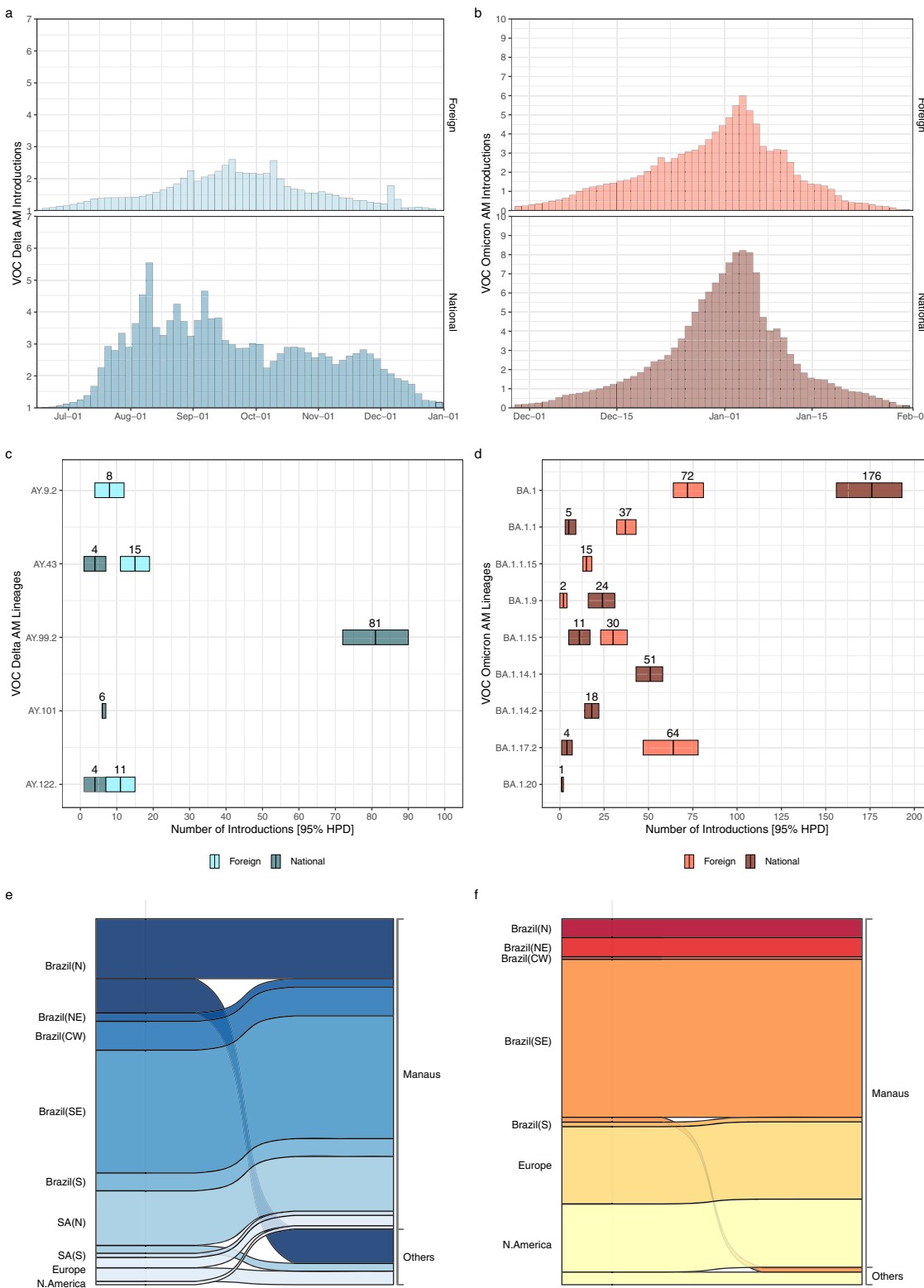

**Fig. 3 | Estimated imports of the most prevalent Delta and Omicron lineages into the Amazonas state. a, b** Estimated number of introductions of Delta and Omicron BA.1 into Amazonas state through time from other countries (foreign) or other Brazilian states (national), based on the Markov jump history in the Bayesian phylogeographic analysis of the most prevalent Delta and Omicron lineages combined. **c, d** Median (and 95% HPD intervals) estimated number of Delta and Omicron

BA.1 introductions into Amazonas state (foreign or national) by lineage. **e, f** Alluvial plots reflect the total estimated number of Delta and Omicron BA.1 imports from each source location (left side) into Manaus or other municipalities (right side), inferred from the Bayesian phylogeographic analyses of all main lineages combined. CW: Central-West. N: North. NE: Northeast. S: South. N: North.

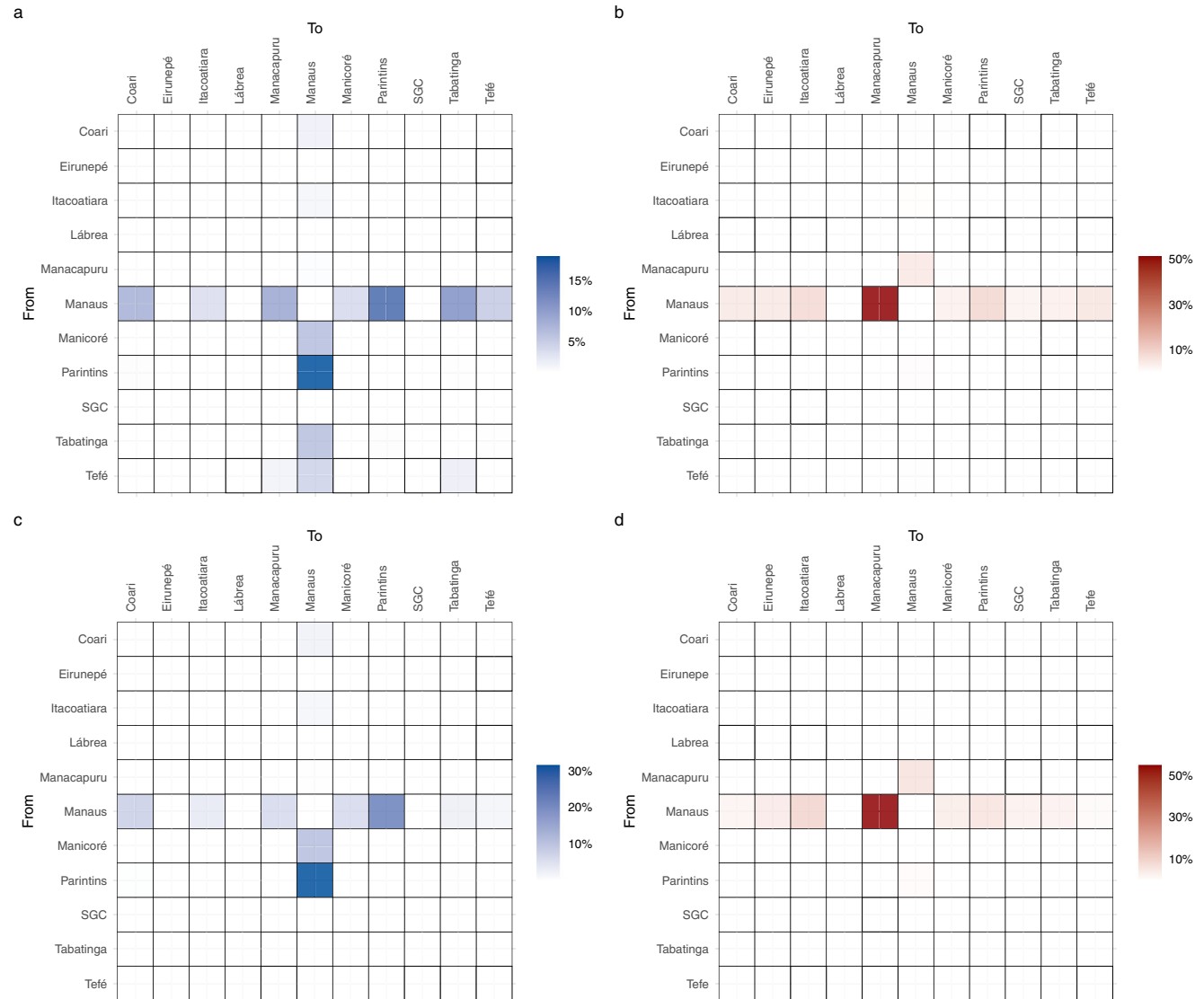

**Fig. 4 | Dissemination of the most prevalent Delta and Omicron lineages within the Amazonas state. a–d** Heatmap cells are colored according to the estimated number of migrations between Amazonian regions for all Delta (**a**) and Omicron (**b**) lineages or for most prevalent lineages AY.99.2 (**c**) and BA.1 (**d**). SGC: São Gabriel da Cachoeira.

interval = 1–2) introduction for lineage BA.1.20 to 246 (95% HPD interval = 227–264) introductions for lineage BA.1 (Fig. 3c, d). Other Brazilian regions (and particularly the Southeastern region) were pointed as the most important sources of Delta (73%) and Omicron BA.1 (46%) viruses introduced into Amazonas, followed by neighboring Amazonian South American countries for Delta (18%) and Europe/North America for Omicron BA.1 (27%/27%) (Fig. 3e, f). The region of Manaus, which comprises the capital city and the major national and international airport of the state, received the most Delta (84%) and Omicron BA.1 (91%) introductions from abroad (Fig. 3e, f).

**Spatial dynamics of Delta and Omicron spread in Amazonas**
Our phylogeographic analysis pointed out that Manaus was the most relevant source of Delta (57%) (Fig. 4a) and Omicron BA.1 (93%) (Fig. 4b) migration events to other Amazonian regions, with a detectable flux connecting the metropolitan region and those containing the most populous municipalities in the state including Parintins, Itacoatiara, Manacapuru, Coari, and Tabatinga. Intra-state viral migrations were not uniformly distributed across lineages. In Delta, lineage AY.99.2 hosted most of the viral movements within the state (59%), followed by lineages AY.9.2 (16%) and AY.43 (12%). In Omicron, most

viral migrations were detected in lineages BA.1 (58%) and BA.1.17.2 (26%). Moreover, the locations most strongly connected also varied across lineages. Within lineage AY.99.2, viral movements were dominated by migration events towards Manaus from Parintins (32%) and Manicoré (11%), and from Manaus to Parintins (22%), Coari (8%), Tabatinga (7%), and Manacapuru (3%) (Fig. 4c). Across the minority Delta lineages analyzed (Fig. S8), Manaus remained as the origin of most viral movements (>50%) in AY.9.2, AY.101, and AY.122, but not in AY.43, where the position was occupied by Tefé. Within lineage BA.1, viral movements were overwhelmingly dominated by migrations events from Manaus to Manacapuru (55%), Itacoatiara (10%), and Parintins (6%) (Fig. 4d). In the minority Omicron lineages analyzed, the metropolitan region was also homogenously pointed out as the primary hub of dissemination in the state (Fig. S9).

**Analysis of the potential impact of geographic sampling bias**
The proportion of SARS-CoV-2 genomes from the Manaus metropolitan region during the period of dominance of Gamma/Delta (80%) was in line with the geographic origin of most SARS-CoV-2 confirmed cases (69%) ($\chi^2$ test, $P > 0.05$) (Fig. 5a). By contrast, the percentage of genomes from Manaus sampled during the Omicron wave (77%) was

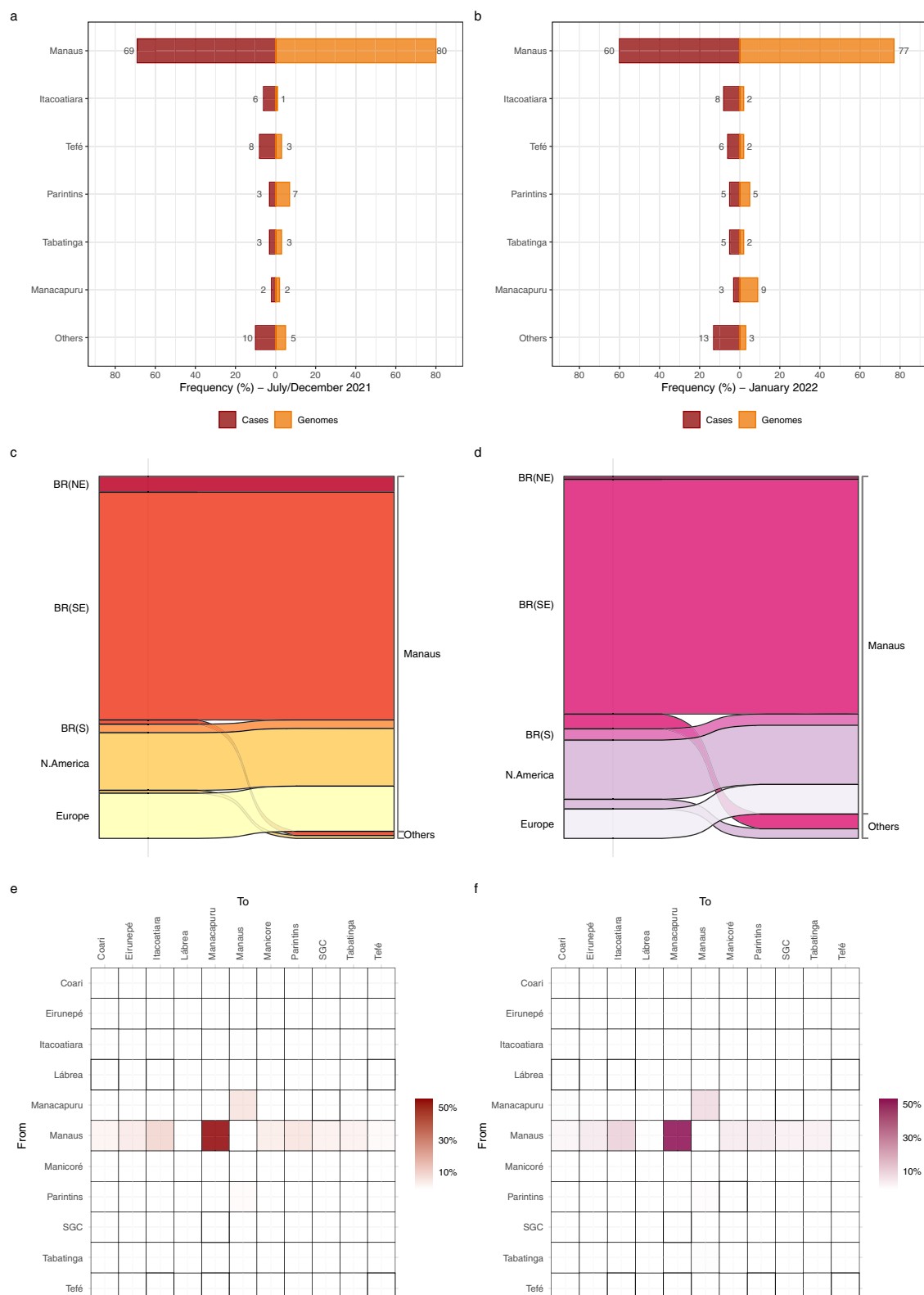

**Fig. 5 | Sampling bias and phylogeographic inferences. a, b** Geographic distribution of SARS-CoV-2 confirmed cases and SARS-CoV-2 genomes from different Amazonian regions during the period of Gamma/Delta (Jul–Dec 2021) (**a**) and Omicron BA.1 (Jan 2022) (**b**). **c, d** Alluvial plots reflect the total estimated number of BA.1 lineage imports from each source location (left side) into Manaus or other municipalities (right side), inferred from the Bayesian phylogeographic analyses with the full dataset (**c**) and a subset proportional to the confirmed number of cases per state region (**d**). **e, f** Heatmap cells are colored according to the estimated number of BA.1 lineage migrations between Amazonian regions inferred from the Bayesian phylogeographic analyses of the full dataset (**e**) and the subset (**f**). BR: Brazil, NE: Northeast, SE: Southeast, S: South, SGC: São Gabriel da Cachoeira.

significantly larger ($\chi^2$ test, $P < 0.05$) than the proportion of SARS-CoV-2 cases detected in that region (61%) (Fig. 5b), probably due to the wider use of rapid antigen tests for SARS-CoV-2 diagnosis (not eligible for genome sequencing) at interior municipalities compared with Manaus. To assess the robustness of our phylogeographic analyses, we performed ancestral reconstructions of the most prevalent BA.1 lineage using phylogenetic trees estimated with the full dataset and with a subset where the proportion of sequences from Manaus was reduced to the corresponding proportion of SARS-CoV-2 cases. Both the full dataset and the subset reconstructions showed that Manaus was the entry point of most BA.1 introductions (96% and 92%, respectively) and the source of most viral migrations to other Amazonian regions (92% and 89%, respectively) (Fig. 5c, f). This indicates that major phylogeographic findings were robust to sampling bias.

**Major Delta/Omicron transmission lineages in Amazonas**

Most viral introductions in Amazonas showed no evidence of significant onward local transmission. Only a few lineages were extensively locally transmitted originating large size transmission

clusters defined as highly supported (aLRT > 0.80) monophyletic clades that descend from a most recent common ancestor (MRCA) probably located (PSP [posterior state probability] > 0.90) in Amazonas and that comprises at least 1% of all Delta ($n > 10$) and Omicron BA.1 ($n > 20$) Amazonian sequences. We identified 15 Delta and 17 Omicron BA.1 Amazonian clades of large size that combined comprise 71% and 47% of all Delta and Omicron BA.1 sequences from Amazonas here analyzed. Most Delta (60%) and Omicron BA.1 (59%) major Amazonian clades belong to lineages AY.99.2 and BA.1 which were mostly (89%) disseminated from other Brazilian regions (Tables S2 and S3). In contrast, most Delta and Omicron BA.1 major clades belonging to other lineages (77%) arose from viruses imported probably from other countries (Tables S2 and S3). Most (72%) Amazonian clades of large size probably originated in Manaus and were mostly sampled in this city, a few local clades (6%) arose in Manaus but were more frequently sampled in inner regions, and the remaining clades (22%) arose and were mainly sampled at inner regions of the state (Fig. 6a, b). Manaus was the most probable entry point of all major Delta and Omicron BA.1 lineages introduced from other countries. The lineage AY.99.2AM-IV

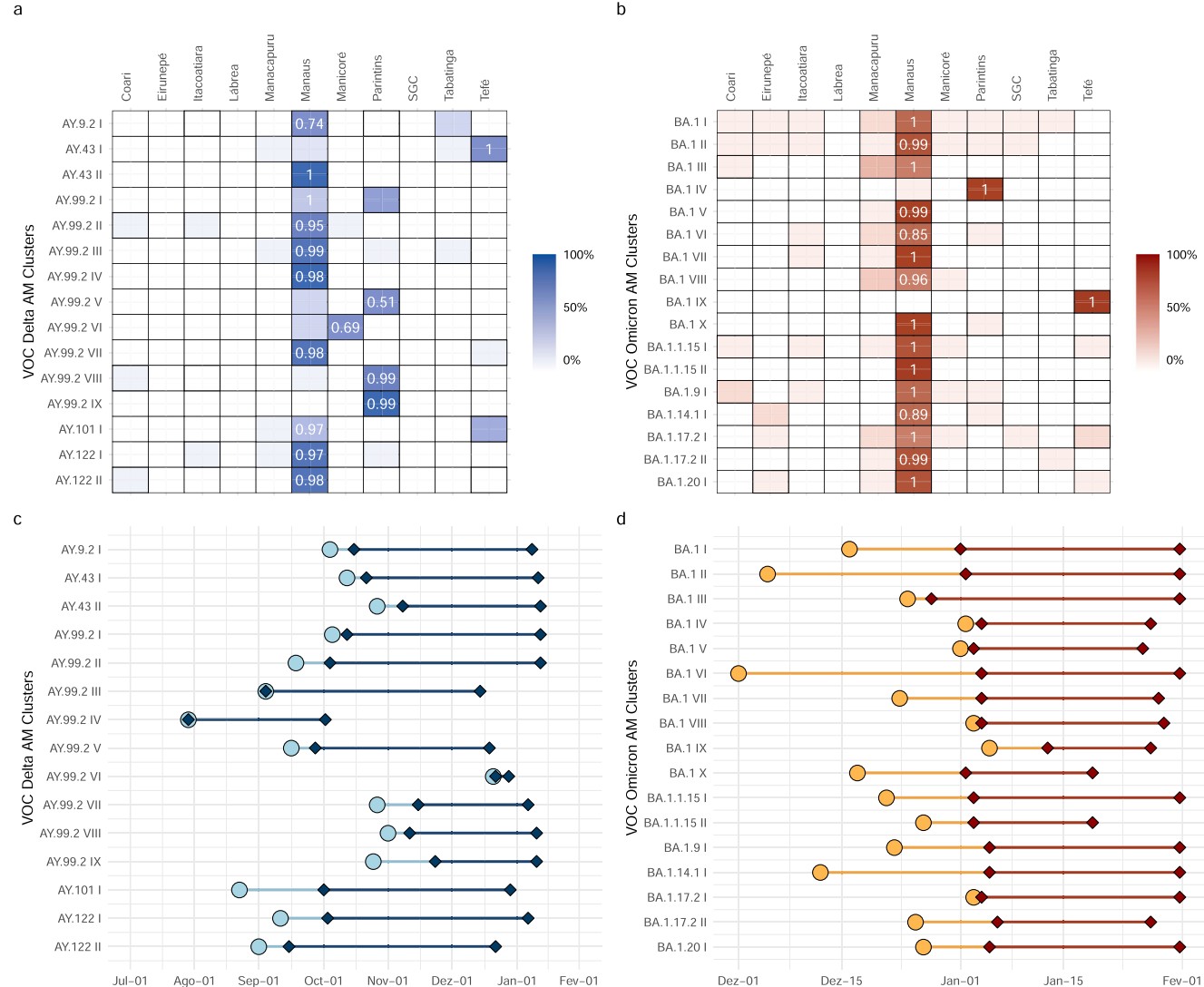

**Fig. 6 | Identification of major Delta and Omicron BA.1 transmission lineages in Amazonas state. a, b** Heatmap cells are colored according to the proportion of Delta and Omicron sequences from different state regions within each major Amazonian transmission lineage. Numbers indicate the posterior state probability (*PSP*) of the most probable source location. **c, d** Timeline of major Amazonian Delta and Omicron genomic clusters. Median $T_{MRCA}$ (circles) and dates of first and last sampling times (diamonds) are plotted on the X axis and genomic cluster on the Y axis. Light color lines correspond to detection lag (i.e., the difference between the $T_{MRCA}$ and the date at which a transmission lineage was detected) and dark color lines correspond to the full sampling range of each transmission cluster.

was the first to become established in Amazonas around late July 2021, but this clade remained restricted to the capital city of Manaus and was last detected in the first week of October 2021. The other major Amazonian Delta lineages only arose after late August 2021 and displayed sustained community transmission until December 2021 to January 2022 (Fig. 6c). All major Amazonian Omicron BA.1 lineages probably originated between early December 2021 and early January 2022 and displayed sustained transmission until the last week of January analyzed (Fig. 6d). Distinct between the two VOCs, Delta's clusters had a mean sampling range (elapsed time between first and last detection) around three times larger (80 days, SD = 18), than Omicron's (25 days, SD = 5). The mean lag in detection since lineage emergence (time of cryptic transmission), however, was quite similar, estimated at 13 days (SD = 10) for Delta lineages and 12 days (SD = 10) for Omicron BA.1 lineages (Fig. 6c, d).

## Effective reproductive number of SARS-CoV-2 VOCs in Amazonas

The relative $R_e$ of VOCs co-circulating in the Amazonas state was first estimated from the observed frequencies of Gamma, Delta, and Omicron from July 2021 to January 2022 using the renewal-equation-based model developed by Ito et al.[26]. The estimated frequencies suggested

that Delta replaced Gamma in September 2021, Omicron BA.1 replaced Delta at the end of December 2021 (Fig. 7a), and the replacement from Delta to Omicron BA.1 resulted in a sudden increase in the average of relative $R_e$ of SARS-CoV-2 infections with respect to (w.r.t.) Delta in this region (Fig. 7b). The relative $R_e$ of Gamma and Omicron BA.1 w.r.t. Delta was estimated to be 0.76 (95% CI: 0.74–0.77) and 3.25 (95% CI: 2.88–3.29), respectively. These results mean that, under the same epidemiological conditions, the average $R_e$ of Omicron BA.1 was 3.25 times higher than that of Delta, the average $R_e$ of the Delta was 1.32 times higher than that of Gamma, and the average $R_e$ of Omicron BA.1 was 4.29 times higher than that of Gamma.

We next used the birth-death skyline (BDSKY) model to compare the $R_e$ of major SARS-CoV-2 Gamma, Delta, and Omicron lineages circulating in Amazonas. To exclude the potential influence of geographical sampling bias, we selected those major Amazonian clusters that originated in the Manaus region, and we only included sequences from the capital city. Although the 95% HPD intervals were relatively large, the trajectory of the median Re values was consistent with each lineage's relative prevalence changes (Fig. 7c). The Gamma lineages displayed a median $R_e$ ~1 in July and <1 from early August onwards. The Delta lineage AY.99.2$_{AM-IV}$ displayed a median $R_e$ > 1 in August and then rapidly fell below one in September 2021, preceding subsequent clade

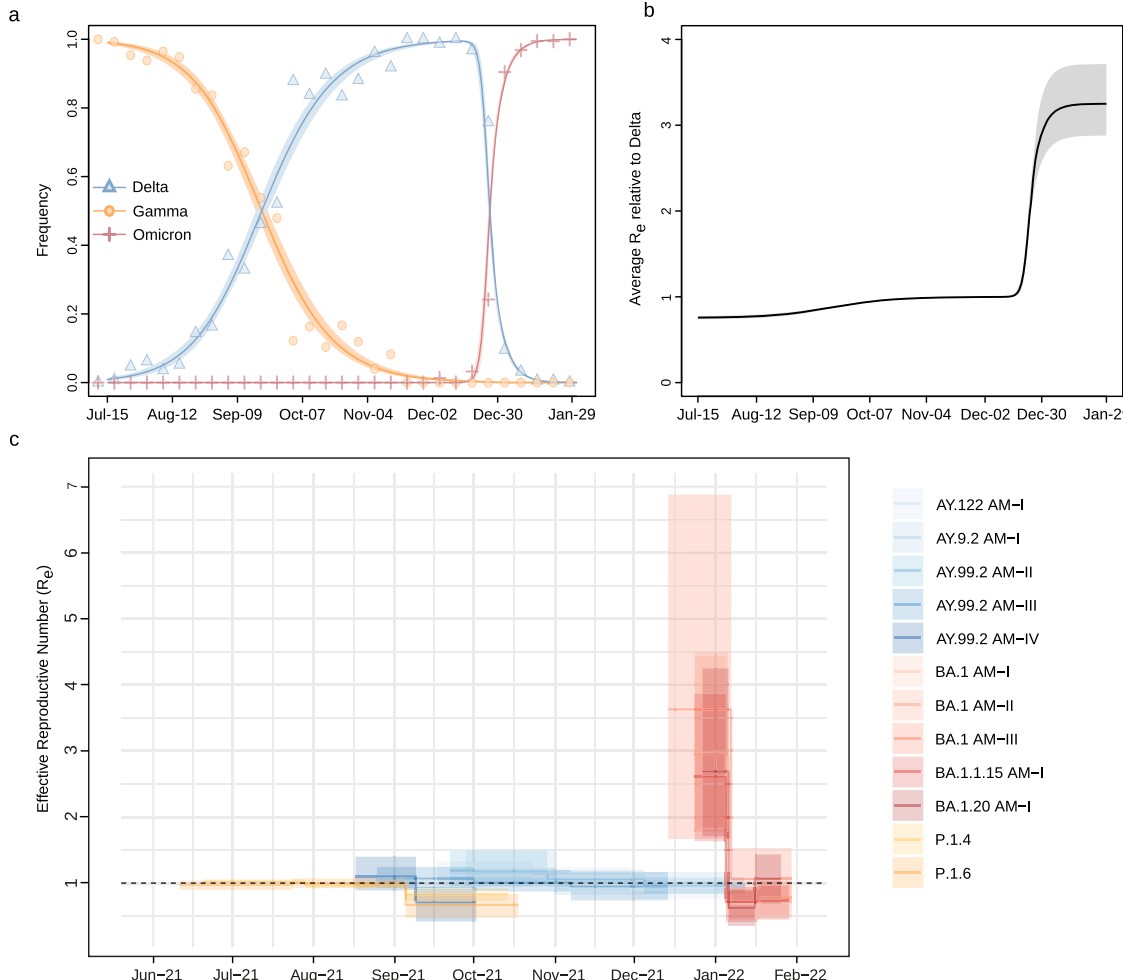

**Fig. 7 | Relative $R_e$ of SARS-CoV-2 VOCs in Amazonas. a** Observed and estimated frequencies of SARS-CoV-2 variants in the Amazonas state. Circles, triangles, and plus signs represent weekly observed frequencies of Gamma, Delta, and Omicron BA-1, respectively. Red, blue, and orange solid lines are estimated average frequencies of Gamma, Delta, and Omicron, respectively, and paled-colored areas represent their 95% confidence intervals. **b** The estimated average relative $R_e$ of SARS-CoV-2 infections with respect to Delta. Solid lines represent the maximum likelihood estimates, and dotted lines represent their 95% confidence intervals (CI). **c** Temporal variation in the Re of major SARS-CoV-2 clades circulating in Manaus, estimated using the BDSKY approach. The estimates' median (solid lines) and 95% HPD intervals (shaded areas) are represented.

extinction. The other Delta lineages (AY.99.2$_{AM-I}$, AY.99.2$_{AM-II}$, AY.99.2$_{AM-III}$, and AY.122$_{AM}$) displayed a median $R_e > 1$ between September and October 2021 and then declined to ≤1 from November onwards. The Omicron BA.1 lineages displayed a high median $R_e > 2$ during the first weeks after emergence but then rapidly declined to <1 from the second half of January 2022. During the periods of co-circulation of major lineages of different VOCs, the average ratio $R_e$ (Delta)/$R_e$ (Gamma) varied between 1.13 and 1.53, and the average ratio $R_e$ (Omicron BA.1)/$R_e$ (Delta) ranged from 2.64 to 3.99.

## Discussion

Our analyses revealed that VOCs Delta and Omicron BA.1 displayed divergent epidemic impacts while spreading in Amazonas. The spread of Delta lineages in Amazonas in the second half of 2021 resulted in a gradual replacement of Gamma lineages without an upsurge of SARS-CoV-2 cases, which continued being detected at a roughly steady-state level from July to December 2021, and a constant CFR. The spread of Omicron BA.1 lineages in Amazonas since late 2021, by contrast, resulted in a rapid replacement of Delta lineages, the exponential growth of SARS-CoV-2 cases, and a substantial reduction in the CFR. These patterns observed are entirely consistent with the predicted differential impact of viral variants with enhanced intrinsic transmissibility (like Delta) or viral variants with combined increased transmissibility and immune escape (like Omicron BA.1) spreading in a well-immunized population in the absence of NPIs[7]. Indeed, analyses of data recovered from the Google COVID-19 Community Mobility Reports revealed that very few restrictions to human mobility were implemented in Amazonas during the period of dissemination of both VOCs Delta and Omicron BA.1 (Fig. S10).

Despite divergent population impacts, our analyses revealed that similar phylogeographic patterns characterized Delta and Omicron BA.1 epidemics in Amazonas. Phylogeographic reconstructions revealed that Delta and Omicron BA.1 epidemics derived from multiple introduction events of different lineages. These introductions resulted in a few successful local transmission chains of a considerable size that, combined, comprise 71% and 47% of all Delta and Omicron BA.1 sequences from Amazonas here analyzed, respectively. The number of local large-size transmission chains of Delta ($n = 15$) and Omicron BA.1 ($n = 17$) variants here identified, and the mean lag in detection (~10 days) were roughly comparable. Introductions of both VOCs in Amazonas mostly resulted from domestic viral migration events, followed by migrations from neighboring countries (Peru, Colombia, and Suriname) for Delta lineages and from non-neighboring regions (Europe and North America) for Omicron BA.1 lineages. The metropolitan region of Manaus acted as the primary entry point and hub of dissemination of both Delta and Omicron BA.1 variants within Amazonas. Moreover, no significant changes in human mobility were detected during the period of dissemination of Delta and Omicron BA.1 in Amazonas (Fig. S10). Thus, the observed differences between Delta and Omicron epidemics in Amazonas were probably driven by viral characteristics rather than human or stochastic factors.

The region of Manaus was pointed out as the main entry point and a major hub of dissemination of Delta and Omicron BA.1 variants within the state, a finding consistent with the central position of Manaus in the human mobility network of the Amazonas state. Manaus is the political, financial, commercial, educational, and cultural center of Amazonas, comprising around 64% of the population of the state. Movements out of Manaus, nonetheless, represented a larger proportion of exporting migrations events for Omicron BA.1 (93%) than for Delta (57%), in line with the reconstructed phylogeographic history of the main transmission clusters of these lineages in the state. In Omicron BA.1, most (88%) detected networks began their dissemination in Amazonas by Manaus, whilst a smaller fraction (60%) did so in Delta. Other relevant primary hubs in Delta, such as Parintins (19%), Tefé (8%), and Manicoré (7%), also coincide with the origin of multiple Omicron clusters. The spatial dissemination dynamics of major Delta and Omicron BA.1 lineages also revealed some differences regarding the viral flux between Amazonas regions. Most viral migrations within Delta lineage AY.99.2 (54%) were between the two most populated regions in the state, Manaus and Parintins. By contrast, most viral migrations within the Omicron lineage BA.1 were from Manaus to Manacapuru (55%) that were located only 71 km apart, followed by migrations from Manaus to Itacoatiara (10%) located 270 km apart, Manaus to Parintins (6%) located 369 km apart, and Manaus to Tabatinga (3%) located 1.106 km apart. These findings suggest that the early spread of lineage BA.1 followed a geographic distance gradient from Manaus, while the dissemination pattern of the long-established lineage AY.99.2 was more complex and probably shaped by other factors than geographic proximity to Manaus.

Two successive VOCs replacements occurred during the study period in the Amazonas state. Firstly, Gamma lineages P.1.4/P.1.6 were substituted by Delta lineages and then Omicron BA.1 lineages replaced Delta. Both replacements seem to have been driven by increased virus transmissibility. Using epidemiological and phylodynamics models, we estimate that the average Re of Omicron BA.1 was ~3 times higher than that of Delta and the average $R_e$ of the Delta was ~1.3 times higher than that of Gamma in Amazonas. These results are consistent with previous findings that support the higher infectiousness of Delta compared to Gamma[27,28] and of Omicron BA.1 compared to Delta[27,29–31]. Together, these data explain the much faster lineage replacement of Delta by Omicron BA.1 than that of Gamma by Delta and support that the relative Re under the same conditions predicts the time course of SARS-CoV-2 lineage replacements[32]. Our results are the first direct comparison between the relative transmissibility of Delta w.r.t. Gamma sub-lineages that harbor the mutations N679K (P.1.4) and P681H (P.1.6). These mutations are particularly interesting because they are located close to the furin cleavage site and probably increase viral infectivity[33]. Interestingly, the relative average Re of lineages P.1.4/P.1.6 w.r.t. lineage P.1 (1.2–1.4)[33], was similar to the estimated relative $R_e$ of Delta w.r.t lineages P.1.4/P.1.6.

While viral characteristics were the primary determinant of the transmissibility of major Delta and Omicron transmission lineages in Amazonas, other factors have also shaped the dynamics of local clade AY.99.2$_{AM-IV}$. We observed that the Amazonian clade AY.99.2$_{AM-IV}$ was the first large Delta local transmission chain established in Manaus in late July 2021 and displayed a median Re in August that was 1.1 times higher than that of Gamma lineages P.1.4/P.1.6. During September 2021, however, the $R_e$ of clade AY.99.2$_{AM-IV}$ fell below one approaching the $R_e$ of lineages P.1.4/P.1.6, and became extinct in early October 2021. This pattern sharply contrasts with the other large Delta transmission chains that later dominated the epidemic in Manaus and that displayed a median $R_e > 1$ during September and October 2021. These findings revealed that different transmission chains of the same SARS-CoV-2 Delta AY.99.2 sub-lineage that co-circulated in Manaus displayed variable epidemic expansion and extinction dynamics. Understanding the factors contributing to the early containment and elimination of lineage AY.99.2$_{AM-IV}$ in Manaus deserves further investigation.

Although Delta replaced Gamma lineages P.1.4/P.1.6 in Amazonas, such variant turnover occurred without a significant upsurge of SARS-CoV-2 cases. This pattern contrasts with the exponential surge of SARS-CoV-2 cases due to the spread of Delta in Europe, North America, Africa, and Asia[16,17,34–38]; but it coincides with the pattern observed in other Brazilian states and South American countries[39,40]. A large fraction of the Amazonian population was probably infected by SARS-CoV-2 (≥70%)[21,22,41,42] and further received at least one dose of the SARS-CoV-2 vaccine (~50%) previous to the onset of the Delta epidemic (Fig. 2a). It suggest that such a high level of hybrid immunity might have prevented the exponential spread of the highly transmissible VOC Delta in Amazonas, and this may explain why the mean Delta's $R_e$ in Amazonas (1.07–1.21) was much lower than the mean Delta's

$R_0$ (5.08) previously estimated[43]. These findings also suggest that the conditional herd immunity reached by South American countries around mid-2021 not only controlled the spread of variants Gamma and Lambda[44] but also protect populations against Delta up-surge. This supports the notion that the population impact of Delta varies across regions depending on underlying population immunological attributes[45].

Hybrid immunity provides greater and more sustained protection against Omicron BA.1 infection than infection or vaccination alone[14,15]. The high percentage of people naturally infected (≥70%) and vaccinated (~65%) (Fig. 2a) in Amazonas in December 2021, however, was not able to prevent the exponential spread of the immune-evasive Omicron BA.1 variant. This observation is consistent with a recent study showing that even individuals with hybrid immunity had low levels of protection against infection by Omicron BA.1 in Brazil[46]. This result also confirms that population immunity acquired by natural infection (before Delta circulation) and/or vaccination might be efficient against Delta, but not against Omicron BA.1[47]. Of note, the average relative $R_e$ of Omicron BA.1 w.r.t. Delta in Amazonas (~3) was roughly comparable to that estimated in other settings with different infection and vaccination histories like South Africa, Denmark, the UK, the United States of America, and India (~2–5)[27,29–31]. The Omicron BA.1 outbreak in Amazonas also had a similar duration from the onset of case growth to their peaks (~30 days) than that observed in other populations with different immune landscapes[18,48,49]. These data support that immunological-related covariates alone could not explain heterogeneity in Omicron's fitness across countries[5,49].

Currently, it is unclear if the higher transmissibility of Omicron BA.1 w.r.t. to Delta is only mediated by its higher ability to infect individuals with prior immunity to SARS-CoV-2 or is also driven by its higher intrinsic transmissibility. Moderate to high levels of immune evasion, without an additional increase in transmissibility, could explain the observed growth advantage of the Omicron BA.1 w.r.t. Delta in different countries like South Africa, the UK, Denmark, the USA, and Australia[3,50]. A previous UK study estimated that Omicron BA.1 is 5–10% less transmissible under a high immune escape scenario than Delta[51]. Interestingly, the median estimated $R_e$ of Omicron BA.1 in Amazonas ($R_e$ = 2.53–2.99) and other countries like India ($R_e$ = 2.33), South Africa ($R_e$ = 2.74–2.79), and the UK ($R_e$ = 3.7–4.0)[28] was lower than the mean estimated Delta's $R_0$ (~5)[43], supporting that the immune evasiveness may primarily drive the rapid spread of Omicron BA.1. Previous immunity, however, confers some level of protection against Omicron BA.1 infection[15] and the actual variant $R_0$ should be thus somewhat higher than the estimated $R_e$.

The higher intrinsic transmissibility and/or immune evasiveness of emergent VOCs surpassed the increasing population immunity levels and caused new COVID-19 waves in Amazonas. Notably, the median $R_e$ of the Omicron BA.1 (2.5–3.0) was very similar to the median $R_e$ of the B.1.* ($R_e$ = 2.1–2.6) and Gamma ($R_e$ = 2.6) variants spreading during the first and second COVID-19 epidemic waves in Amazonas, respectively[20]. The most recent COVID-19 wave in Amazonas, however, displayed some unique features. The Omicron BA.1 wave displayed a higher peak of SARS-CoV-2 cases and shorter duration than previous waves, as observed in other Western countries[18,48,49]. We speculate that the weak mitigation measures implemented in Amazonas during the dissemination of Omicron BA.1 may have resulted in a high and sharp peak with a rapid decline, while the curve of previous variants may have been "flattened" and prolonged by more stringent control measures implemented earlier (Fig. S10). Moreover, the Omicron BA.1 wave in Amazonas displayed a lower CFR than previous waves as also observed in other settings[52–55], which could be explained by extensive acquired population immunity combined with lower intrinsic viral pathogenicity. Of note, the CRF during the Delta dominant period was higher than during the Omicron BA.1 dominant wave,

consistent with the notion that Delta is intrinsically more virulent than Omicron BA.1[52,53,56].

This work has some limitations that should be taken into consideration. First, discrete phylogeographic reconstructions may have been impacted by uneven sampling across geographical locations. Our sequence data represent a random sample of all SARS-CoV-2 positive cases detected in private and public health institutions from the entire Amazonas state and there was no intended bias considering dense sampling of specific outbreaks in selected locations. However, the selection of samples with diagnosis by RT-PCR reduces the proportion of sequences from interior municipalities where a high proportion of SARS-CoV-2 diagnoses were performed by rapid antigen tests, particularly during the Omicron epidemic. Second, the inferred number of viral imports into Amazonas could have been affected by sampling coverage and the power to identify transmission clusters from genomic data of variants with extremely rapid global dissemination and low regional diversification. We sequenced only a small fraction of infected individuals, and viral imports should thus be interpreted as lower-bound estimates. On the other hand, inferred genomic transmission clusters and some "unclustered" sequences may be part of larger epidemiological transmission clusters, and viral imports may have been thus overestimated. Third, phylogenomic estimates of Re displayed large credibility intervals, making it challenging to obtain precise estimations about relative viral variants' transmissibility. Fourth, we cannot formally test the links between hybrid immunity and the divergent trajectory of Delta and Omicron epidemics in Amazonas because of the lack of data about the vaccination status of individuals sequenced in this study.

In summary, our findings support that the dynamics and population impact of SARS-CoV-2 lineage replacements in Amazonas during the second half of 2021 varied according to the virological characteristics. The introduction of Delta lineages with high intrinsic transmissibility resulted in a gradual replacement of Gamma lineages without an upsurge of SARS-CoV-2 cases. Introduction of the immune-evasive Omicron BA.1 lineages, by contrast, resulted in a very fast replacement of Delta lineages and a large upsurge of SARS-CoV-2 cases. We propose that the high level of hybrid immunity (infection plus vaccination) acquired by the Amazonian population by mid-2021 effectively reduced the transmissibility of Delta lineages ($R_e$ = 1.1–1.2), but was not able to prevent the rapid spread of the Omicron BA.1 lineages ($R_e$ = 2.0–3.0). Previous immunity, however, was probably crucial to decouple deaths from SARS-CoV-2 infections during the Omicron wave in Amazonas. Therefore, optimization of vaccination strategies will be needed to curtail the transmission of future immune-escape viral variants in populations with high levels of hybrid immunity.

## Methods

### SARS-CoV-2 samples and ethical aspects

A total of 4128 SARS-CoV-2 positive samples with real-time RT-PCR cycling threshold ($C_t$) below 30, collected between July 1, 2021 and January 31, 2022, from residents of 48 out of 62 municipalities from the Amazonas state, were randomly selected for genome sequencing. Nasopharyngeal swabs (NPS) samples collected from suspect COVID-19 cases by sentinel hospitals or healthcare units were positively tested by real-time RT–PCR with the commercial assay: SARS-CoV2 (E/RP) (Biomanguinhos, https://www.bio.fiocruz.br/index.php/br/ produtos/reativos/testes-moleculares/novocoronavirus-sars-cov2/ kit-molecular-sars-cov-2-e-rp). The Amazonas State Health Surveillance Foundation - Dra. Rosemary Costa Pinto (FVS-RCP/AM) and the Central Laboratory from the State of Amazonas (LACEN-AM) sent SARS-CoV-2 positive samples for sequencing at FIOCRUZ Amazônia, part of the local health genomics network (REGESAM) and the FIO-CRUZ COVID-19 Genomic Surveillance Network. The FVS-RCP/LACEN-AM has been the principal responsible for the molecular diagnosis of SARS-CoV-2 since the very beginning of the COVID-19 epidemic in the

Amazonas state and receives suspected samples from private and public health institutions from the entire state. Initially, all positive samples with a real-time RT-PCR cycling threshold ($C_t$) below 30 were qualified for sequencing. Next, we selected a random subset of positive samples by epidemiological week and municipality, avoiding overrepresenting one local to the detriment of the others. There was no intended bias considering dense sampling of specific outbreaks and we also do not consider the health status (severe, mild, or asymptomatic) or epidemiological data (sex and age) of individuals as a criterion for sampling. Thus, our sequence data represent a random sample of all SARS-CoV-2 positive samples detected in the Amazonas state in the study period with no bias other than diagnosis by real-time RT-PCR and $C_t < 30$. This study was conducted at the request of the SARS-CoV-2 surveillance program of FVS-RCP/AM. It was approved by the Ethics Committee of Amazonas State University (n°. 25430719.6.0000.5016), which waived signed informed consent.

### SARS-CoV-2 amplification, library preparation, and sequencing

The viral RNA was subjected to reverse transcription (RT) and PCR amplification using the Illumina COVIDSeq Test (Illumina), including some primers to cover regions with dropout[33]. Normalized pooled amplicons of each sample were used to prepare NGS libraries following manufacturers' instructions and sequenced in paired-end runs on Illumina MiSeq or NextSeq 1000 platforms at Fiocruz Amazônia. Raw data was collected using MiSeq Control Software 2.6.2.1 or NextSeq 1000/2000 Control Software Suite v1.2.0.

### SARS-CoV-2 whole-genome consensus sequences and genotyping

Raw data were converted to FASTQ files at Illumina BaseSpace cloud (https://basespace.illumina.com) and consensus sequences were produced with DRAGEN COVID LINEAGE 3.5.5 or 3.5.6. Genomes were evaluated for mutation calling and quality with Nextclade Web 1.13.0 (https://clades.nextstrain.org)[57]. Sequences with more than 3% of ambiguities "Ns" or any quality flag were reassembled using a customized workflow that employed BBDuk and BBMap tools (v38.84) running on Geneious Prime 2022.0.1. Our consensus sequences had a mean depth coverage higher than 1800x, excluding duplicate reads. Only those considered high-quality were considered for further analysis. Whole-genome consensus sequences were classified using the 'Phylogenetic Assignment of Named Global Outbreak Lineages' (PANGOLIN) software v3.1.18 (pangolearn 2022-01-20, constellations v0.1.2, scorpio v0.3.16, and pango-designation release v1.2.123)[58] and later confirmed using phylogenetic analyses.

### Selection of SARS-CoV-2 global reference sequences

To infer introductions for each major SARS-CoV-2 Delta and Omicron lineages circulating in Amazonas, we retrieve all complete SARS-CoV-2 genomes from around the world from the EpiCoV database in GISAID (https://www.gisaid.org/)[59] from the respective lineages covering the same time frame than Amazonian sequences (July 1, 2021–January 31, 2022). To select the most closely related reference sequences to the ones in our datasets, a variable number of those with the highest similarity score under a local BLAST v2.13.0[60] search was selected. In all cases, we made sure that reference sequences represented the largest fraction of the final dataset (~75%). This strategy aimed to keep the total number of sequences within a computationally tractable range (<4000), but also keep the necessary genomic diversity to discriminate potential dissemination networks.

### Maximum likelihood phylogenetic trees and temporal signal

SARS-CoV-2 Delta and Omicron complete genome sequences from Amazonas were aligned with the global reference datasets of the corresponding lineage using MAFFT v7.467[61] and subject to maximum likelihood (ML) phylogenetic analysis using IQ-TREE v2.1.2[62] under the general time-reversible (GTR) model of nucleotide substitution with a gamma-distributed rate variation among sites, four rate categories (G4), a proportion of invariable sites (I) and empirical base frequencies (F) nucleotide substitution model, as selected by the ModelFinder application embedded in IQ-TREE[63]. The approximate likelihood-ratio test assessed the branch support based on the Shimodaira–Hasegawa-like procedure (SH-aLRT) with 1000 replicates. The temporal signal of the Delta and Omicron assembled datasets was assessed from the ML tree by performing a regression analysis of the root-to-tip divergence against sampling time using TempEst v1.5.36[64]. Preliminary analysis showed minimal temporal signal in most datasets, so all evolutionary analyses (see below) were performed using an informative uniform prior to the molecular clock rate ($5-15 \times 10^{-4}$ substitutions/site/year) that covers the mean substitution rates obtained in earlier studies across different time-scales[65–67].

### Estimating time-scaled phylogenetic trees

The time-scaled phylogenetic trees of Delta and Omicron Amazonian transmission clusters were estimated in BEAST v1.10[68] under a GTR + F + I + G4 nucleotide substitution model, a strict molecular clock model, and a non-parametric Bayesian skyline (BSKL) model as the coalescent tree prior[69]. Due to the large size of Delta and Omicron lineages datasets ($n = 372-3445$ sequences), time-scaled trees were reconstructed using a modified version of BEAST (https://beast.community/thorney_beast) that makes feasible the analysis of big datasets. The method alleviates most of the computational burden by fixing the tree topology and then re-scales the branch lengths based on the clock and coalescent models. The ML phylogenetic trees were inputted in the BEAST XML file as starting, and data trees and analyses were performed as specified above. MCMC was run sufficiently long to ensure convergence (effective sample size > 200) in all parameter estimates as assessed in TRACER v1.7[70]. The maximum clade credibility (MCC) trees were summarized with TreeAnnotator v1.10 and visualized using FigTree v1.4.4 (https://github.com/rambaut/figtree/releases). XML files used in the temporal analysis are available at https://github.com/flaviviruslab/covid_am_delta_omicron.

### Discrete Bayesian phylogeographic analyses

A set of 1000 trees was randomly selected from the posterior distribution of trees resulting from the BEAST analysis of the Delta and Omicron entire lineages datasets. Sampling locations were used as traits in the phylogeographic model, and the ancestral states were reconstructed using a discrete symmetric model[71]. XML files used in the phylogeographic analysis are available at https://github.com/flaviviruslab/covid_am_delta_omicron. Samples used were designated to discrete locations ($n_{TOTAL} = 27$), selected to balance both the need for a detailed reconstruction of the multiple transmission pathways that led to the introductions of VOCs Delta and Omicron in Amazonas and also to limit the total number of parameters to control the computational time demand. Geographical labels were selected with increased detail as one moves closer to the Amazonas state. Foreign samples were grouped by continental and sub-continental regions ($n = 11$), Brazilian sequences were grouped by geo-political regions ($n = 5$), and sequences from Amazonas by sub-state regions ($n = 11$). We complemented this analysis with a Markov jump estimation of the number of location transitions throughout evolutionary history[72], which was used to explore the directionality of the transitions[71]. Viral migrations within Amazonas were estimated as transitions between Amazonian locations in the inferred phylogeographic trees. Major genomic transmission clusters were defined as highly supported (aLRT > 0.80) monophyletic clades that descend from an MRCA node probably located (PSP > 0.90) in Amazonas and that comprise at least 1% of all Delta ($n > 10$) and Omicron ($n > 20$) Amazonian sequences here analyzed. The temporal history of Delta and Omicron imports into Amazonas was summarized as histograms in one-week periods.

Supported transitions into Amazonian locations from non-Amazonian ones and transitions among these Amazonian locations were visualized, respectively, in alluvial plots generated with the package ggaluvial v0.12.4[73], and heatmaps generated with ggplot2 v3.4.0[74], both available in R language (https://www.r-project.org).

### Effective reproductive number estimations

The relative $R_e$ of viral variants in Amazonas was estimated from the observed frequencies of Gamma, Delta, and Omicron from July 2021 to January 2022 using the renewal-equation-based model developed by Ito et al.[26]. The generation time of Gamma and Delta were assumed to follow a gamma distribution having a mean of 4.7 and a standard deviation of 3.3[75]. The generation time of Omicron was assumed to follow a gamma distribution having a mean of 2.8 and a standard deviation of 1.98[30]. We use Delta as the baseline variant to calculate the relative $R_e$ of Gamma and Omicron. The relative $R_e$ of Gamma and Omicron with respect to (w.r.t.) Delta was estimated by maximizing the multinomial likelihood of the variant frequencies calculated by the model. The 95% confidence intervals of estimated parameters were calculated by the profile likelihood method[76]. The average of relative generation times and relative $R_e$ were calculated by multiplying the frequency and the relative $R_e$ of each variant w.r.t. Delta and summing them up[32]. The temporal $R_e$ trajectories of Gamma, Delta, and Omicron variants in Manaus were estimated from genomic data by using the BDSKY model[77] implemented within BEAST 2 v2.6.2[78]. We selected all Amazonian transmission clusters that probably arose in the capital city and comprised at least 40 sequences after removing those sampled outside Manaus. The sampling rate was set to zero for the period before the oldest sample and then estimated from the data afterward. The BDSKY prior settings were as follows: become uninfectious rate (exponential, mean = 36); reproductive number (lognormal, mean = 0.8, s.d. = 0.5); sampling proportion (beta, alpha = 1, beta = 100). The origin parameter was conditioned to root height, and Re was estimated in a piecewise manner over 3–6 time intervals defined from the date of the most recent sample up to the root of the tree. For each transmission cluster tMRCA we applied a uniform prior whose limits were based on the 95%HPD of the estimated origin according to the phylogeographic analysis. MCMC chains were run until all relevant parameters reached ESS > 200, as explained above.

### Human mobility data from Amazonas

We obtained daily human mobility data in Amazonas from Google COVID-19 Community Mobility Reports[79], which uses aggregated and anonymized mobility data to measure the change in total visitors compared to a baseline value at different categories of places over time. We extracted data aggregated across Amazonas state between January 15, 2020 and March 31, 2022, covering the three main SARS-CoV-2 epidemic waves in that state. The moving average was calculated for all mobility trends considering seven-day intervals.

## Data availability

The findings of this study are based on 13,776 sequences both generated and previously published up to March 31, 2022, and accessible at https://doi.org/10.55876/gis8.220913va. Genomes generated in this study are also available in GenBank under accession numbers OQ521669 - OQ525782.

## Code availability

All Beast XML files used in both temporal and phylogeographic analysis are available at https://github.com/flaviviruslab/covid_am_delta_omicron.

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

## Acknowledgements

We gratefully acknowledge all data contributors, i.e., the Authors and their Originating laboratories responsible for obtaining the specimens, and their Submitting laboratories for generating the genetic sequence and metadata and sharing via the GISAID Initiative, on which this research is based. The authors also wish to thank all the healthcare workers and scientists who have worked hard to deal with this pandemic threat. In addition, we appreciate the support of FIOCRUZ COVID-19 Genomics Surveillance Network members and the Respiratory Viruses Genomic Surveillance Network of the General Laboratory Coordination (CGLab) of the Brazilian Ministry of Health (MoH), Brazilian Central Laboratory States (LACENs), and the Amazonas surveillance teams for the partnership in the viral surveillance in Brazil. Funding support FAPEAM (PCTI-EmergeSaude/AM call 005/2020; Rede Genômica de Vigilância em Saúde - REGESAM); Conselho Nacional de Desenvolvimento Científico e Tecnológico (grant 403276/2020-9); Inova Fiocruz/Fundação Oswaldo Cruz (Grant VPPCB-007-FIO-18-2-30 - Geração de Conhecimento); Centers for Disease Control and Prevention (CDC Grant Award 002174); Departamento de Ciência e Tecnologia (DECIT) of the Brazilian MoH; PAHO, Brazilian office and AIDS Healthcare Foundation (AHF - Global Public Health Institute); NPI EXPAND - U.S. Agency for International Development (USAID) implemented by Palladium (7200AA19CA00015). CNPq through their productivity research fellowships (306146/2017-7, 303902/2019-1, 304883/2020-4 and 313403/2018-0, respectively): F.G.N., G.L.W., G.B., and M.M.S., respectively. FAPERJ (Grant number E-26/202.896/2018): G.B. FAPERJ- Fundação Carlos Chagas Filho de Amparo à Pesquisa do Estado do Rio de Janeiro, grant SEI-260003/019669/2022: I.A.

## Author contributions

I.A., G.B., and F.G.N. conceived and designed the study and contributed to data analysis. V.N., V.S., A.S., D.S., F.N., M.M., M.J.B., L.G., and G.S. contributed to diagnostics and sequencing analysis. C.F.C., L.A., J.H.S., and T.C.A.R. contributed to the patient and public health surveillance data. C.P. and K.I. contributed to effective reproductive number (Re) estimations. FGN and MMS contributed to laboratory management and obtaining financial support. P.C.R., G.L.W., E.D., and T.G. contributed to formal data analysis. I.A. and G.B. wrote the first draft and all authors contributed and approved the final manuscript.

## Competing interests

The authors declare no competing interests.
