## [Peer Review File · Nature Communications]

Comparative epidemic expansion of SARS-CoV-2 variants Delta and Omicron in the Brazilian State of AmazonasREVIEWER COMMENTS

Reviewer #1 (Remarks to the Author):

Review of "Comparative epidemic expansion of SARS-CoV-2 variants Delta and Omicron in Amazonas, a Brazilian setting with high levels of hybrid immunity"

What the paper is about / its relation with global context

In this paper, Arantes, Bello and colleagues describe the waves of SARS-CoV-2 epidemics in the Brazilian state of Amazonas. Specifically, they aim at comparing and understanding the trajectories of epidemic waves caused by Delta and Omicron in a setting displaying what they call hybrid immunity (infection plus vaccination). In a global context of SARS-CoV-2 lineage replacement previously driven by intrinsic transmissibility increase and now apparently driven by the emergence of immune-escape variants, a better understanding of the interaction of emerging VoCs with their immunological environment is pivotal to a better control of the SARS-CoV-2 pandemic.

What the authors have done

In this study, Arantes, Bello and colleagues generated a very valuable dataset. Manaus the particular feature of both having experienced a substantial epidemic wave caused by the Gamma VoC and, as the authors emphasized, likely have displayed particularly high immunity levels quite soon in the pandemic. The paper is clear and well written.

The authors rely on prevalence, cases numbers, fatality rate and reproductive number of the three VoC Gamma, Delta and Omicron to evaluate which factors drove the SARS-CoV-2 lineage replacement dynamics in Amazonas during the second half of 2021.

Additionally, a phylogenetic approach was performed in order to establish epidemiological links between Amazonas SARS-CoV-2 isolates and those from other parts of the country and of the world. Claims are made as to the number of introductions and their geographic origin.

Major comments :

The authors use one given epidemiological setting (probably variable in time due to immunological, social etc factors) to compute R_e values of multiple replacing lineages. Then, they state that the R_e values (from identical settings, which is not necessarily true in their case) they computed is a very good explainer of the lineage replacement pattern they observe. There is quite some circularity here.

One important claim of the paper is that in Amazonas, there was no surge in cases linked the Delta VOC (higher intrinsic transmissibility) because immunity levels were high in the population. However, Omicron did produce a surge in cases, linked to its immune escape abilities. The above is probably true, but while values of R_e are formally established, there is not much that backs up the claim mentioned above. For example, there is no formal assessment of the links between hybrid immunity (or lack thereof) and the trajectory of the epidemic waves.

The authors should also mention that on top of being due to previous exposure, the reduced fatality rate linked to Omicron infections might also be due to its (possible) reduced intrinsic virulence.

A substantial amount of results from this study rely on the reliability of case numbers from Cota 2020. Any bias or modification of the sequencing / sampling intensity across time and / or space could have a substantial impact on the case number and hence the epidemic waves' trajectories studied herein. the authors should mention how they ensured that this bias (or lack thereof) was assessed.

Minors comments:

- Fig1 A: y axis contains "?"
- Fig5: in the caption and in the figure, you used R_t instead of R_e
- The use of different nomenclature (while in my opinion unavoidable in this paper) sometimes

makes the reading tricky. Try to be as clear as possible and to make comparisons possible between lineages in one claim / sentence. For example, line 111 "as Omicron (B.1.1.529/BA.1.*) and four (<1%) as B.1.621" you should specify "(Mu)" at the end of the sentence. Check all the manuscript.

- Not sure that the colors of all figures are color blind-friendly.
- Check readability of figure captions according to the size they display for big figures. For example, color of the lines of the legend of fig1 C D E and fig2 D E are difficult to see when watched at 100%. The same goes for the text of figure 4. Check all figures.
- Why isn't there Delta in Fig 2 D and E ?
- Line 181, BA.14.2 should probably read BA.1.14.2
- Specify how is CFR computed in the text and not only in fig2 caption
- Make sure to make the data publicly available if not already done

Reviewer #2 (Remarks to the Author):

Arantes et al generate over 4000 SARS-CoV-2 genomes from the Amazonas region and combine these with globally available genomes and case data to investigate the importation and transmission patterns of SARS-CoV-2 lineages and factors influencing these patterns. The Authors find that both Delta and Omicron (BA.1) variants were imported into Amazonas multiple times. Several dominant transmission lineages arose, mostly in the capital Manaus. The replacement dynamics of Delta and Omicron differed greatly, with Omicron (BA.1) exhibiting higher R_e than Delta and Gamma. This is an excellent manuscript that carries out gold standard analyses to reach well supported conclusions. I have only one suggestion and several minor comments.

In the section starting on line 213 ("Identification of major Delta and Omicron transmission lineages in Amazonas"), the Authors find that Manaus acted as a major source of viral lineages to other regions within Amazonas and most large Amazonian clades likely originated in Manaus. While this makes sense epidemiologically, the majority (~80%) of Amazonas sequences within the study are from Manaus. Therefore to enable robust conclusions about the sources and origins of viral lineages, the Authors should repeat the phylogeographic analyses with random subsamples of Manaus sequences (for example to the number of sequences from the second most common location) and ensure that the same patterns are present.

Minor comments

Line 62 – as the Omicron section of the manuscript is focussed on BA.1, it would be useful to include a description of the major Omicron lineages (BA.1, BA.2, BA.4 and BA.5) and their circulation in the introduction

Figure 1A – y-axis labels show "10?", presumably should be updated

Figure 1A – how many days are included in each rolling date? It's not currently clear whether bars correspond to cases or genomes (presumably cases) – is it possible to clarify in the legend?

Figure 1C – this panel would be easier to interpret if the x-axis was dates (as in 1A) rather than days

Figure 1F doesn't seem to exist, was this merged into 1E?

Figure 2D and 2E – is the Omicron colour correct in the legend?

Lines 159-162 – should state that this is the CFR for detected cases, not for overall cases

Line 178 – the Delta lineages should be B.1.617.2/AY.* rather than B.1.617/AY.*

Lines 193-194 – some more discussion around the number of introductions per lineage would be useful. The number of introductions doesn't seem to be directly comparable between lineages as lineages have different sizes and ranges. For example, BA.1 is a very large globally distributed

lineage so it might be expected to be imported multiple times. Whereas BA.1.14.2 is more localised to Brazil so international imports are expected to be rare

Figure 3A and 3B – it seems like these panels would be more informative in number of introductions rather than proportion of introductions

Figure 3C and 3D – it would be useful to split this into introductions into Amazonas from elsewhere in Brazil and international introductions to show different sources for different lineages

Figure 4A – there seems to be a high number of virus movements from Parintins to Manaus within Delta. Can the Authors include a description of these movements and whether they are spread across transmission lineages? I.e. is that a general pattern across lineages or specific to one lineage?

Line 271/Figure 5 – lineage AY.99.2AM-V doesn't appear to be in Figure 5C, it might be useful to include as it is discussed in the text

Lines 377-378 – the term "population immunity" seems too broad here. Are the Authors referring to immunity acquired before Delta and from vaccines targeted against the original SARS-CoV-2 lineage?

Lines 411-414 – it seems difficult to separate the role of control measures from other factors. There are other countries that imposed restrictions in earlier waves but not with Omicron – do other countries show similar differences in curve shape between waves?

Lines 476-478 – what were the parameters for excluding sequences after this reassembly?

Section starting on line 528 – the discrete labels used in the phylogeographic analysis and the logic behind the choice of labels should be outlined

Line 546 – seems to be a typo in "ggaaluvial" which should be "ggalluvial"

Reviewer #3 (Remarks to the Author):

This manuscript by Arantes and colleagues describes the population-level impact and phylogeographical dynamics of Delta and Omicron in the Amazonas state. This is an area that was one of the most severely affected regions within South America with a very high population infection rate. Thus, it represents a unique point of study for such analysis as demonstrated here. The authors generate over 4,000 high-quality whole genome sequences from SARS-CoV-2 infected individuals in all 11 Amazonas state regions between July 2021 and Jan 2022 which represents approx 3.2% of all lab cases. Overall, I find these findings not very surprising but interesting as they do offer some insights into the Delta and Omicron competition epidemics in a very concentrated region. However, I do think that there are several points where clarity is needed that could help strengthen the manuscript. The biggest question is regarding sampling and how was it performed and the nature of such biases. The authors do cite sampling as a limitation but they merely talk about it from a numbers perspective as oppose to the methodology of sampling: randomly chosen samples vs. dense sampling of specific outbreaks in certain facilities such as hospitals etc. This will have an effect on both the size of the transmission chains and the lag

- How are these samples chosen for sequenced and even sampled. Is it random or is it mainly derived from hospitals. Some clarity on the nature of sampling is required and the limitations of this approach is warranted.
- The lack of temporal signal in the assembled datasets is maybe not surprising given the relatively slow evolutionary rates of SARS-CoV-2 and the short study period. However, it is not clear where the rate of $5-15 \times 10^{-4}$ substitutions/site/year comes from. Can the authors justify this with a reference or further analysis?
- I applaud the authors for using the thorny version of BEAST to carry out an analysis of a bigger

dataset but these xml files should be made publicly available on a github or zendo repository as I think they would be of use to many researchers and the fact that it is the trees and not sequence data that is shared within these xmls there should be no prohibitive effect of sharing from GISAID. In fact given they chose this approach they could have easily scaled their numbers up.

- How many inferred introduction events were singletons? Given the sequencing coverage rate this may also imply that many of these singletons may actually represent small transmission lineages assuming that the rate of undersampling is uniform across transmission lineages of all sizes. In fact the authors should clearly define what constitutes a transmission chain as this varies depending on the phylogenetic study in question.
- Do the authors have any available data on vaccination rates per region and how this may correlated with their findings?
- I'm assuming that there were no restriction in terms of social mixing or curfews etc that could help explain the data during the observed study period.

REVIEWER COMMENTS

REVIEWER #1

MAJOR COMMENTS

The authors use one given epidemiological setting (probably variable in time due to immunological, social etc. factors) to compute R_e values of multiple replacing lineages. Then, they state that the R_e values (from identical settings, which is not necessarily true in their case) they computed is a very good explainer of the lineage replacement pattern they observe. There is quite some circularity here.

1) One important claim of the paper is that in Amazonas, there was no surge in cases linked the Delta VOC (higher intrinsic transmissibility) because immunity levels were high in the population. However, Omicron did produce a surge in cases, linked to its immune escape abilities. The above is probably true, but while values of R_e are formally established, there is not much that backs up the claim mentioned above. For example, there is no formal assessment of the links between hybrid immunity (or lack thereof) and the trajectory of the epidemic waves.

Reply: The reviewer is correct. Unfortunately, we have no data about the vaccination status of individuals sequenced in this study so we cannot formally test the links between levels of hybrid immunity and the divergent trajectory of Delta and Omicron epidemic waves in Amazonas. We modified the manuscript to present hybrid immunity as a hypothetical explanation of the differences observed, and we also included the lack of information about vaccination status as a limitation of our study.

2) The authors should also mention that on top of being due to previous exposure, the reduced fatality rate linked to Omicron infections might also be due to its (possible) reduced intrinsic virulence.

Reply: We modified the discussion to consider both factors (previous immunity and reduced intrinsic virulence) as possible causes of the reduced CFR during the Omicron wave in Amazonas. We also modified the abstract section's last sentence to avoid any specific statement about the putative causes of such a reduction.

3) A substantial amount of results from this study rely on the reliability of case numbers from Cota 2020. Any bias or modification of the sequencing / sampling intensity across time and / or space could have a substantial impact on the case number and hence the epidemic waves' trajectories studied herein. the authors should mention how they ensured that this bias (or lack thereof) was assessed.

Reply: On the one hand, Cota 2020 aggregates data from different official sources including: 1) the official page of the Brazilian Ministry of Health

(<https://covid.saude.gov.br/>) that updates the number of cases and deaths per federative unit once a day, and 2) the official epidemiological bulletins of each federative unit (<https://brasil.io/dataset/covid19/caso/>) that compiled daily data at municipal level. Thus, the epidemic waves trajectories described here reflect the official data. On the other hand, the samples sequenced in this study were received from the Health Surveillance Foundation - Dra. Rosemary Costa Pinto (FVS-RCP/AM) through the Central Laboratory from the State of Amazonas (LACEN-AM). This public health unit has been the principal responsible for the molecular diagnosis of SARS-CoV-2 since the very beginning of the COVID-19 epidemic in the Amazonas state and receives suspected samples from private and public health institutions across the entire state. Initially, all positive samples with a real-time RT-PCR cycling threshold (Ct) below 30 were qualified for sequencing. We randomly selected a subset of positive samples by epidemiological week and municipality to obtain a representative sample across time and space. We also do not consider the health status (severe, mild, asymptomatic) or the epidemiological data (sex, age) of individuals as a criterion for sampling. Thus, our sequence data represent a random sample of all SARS-CoV-2 positive cases detected in the Amazonas state in the study period with no bias other than diagnosis by RT-PCR and Ct values <30. Because rapid antigen tests for SARS-CoV-2 diagnosis were more widely used in countryside municipalities than in the metropolitan region of Manaus, particularly during the Omicron wave, this introduced a slight geographical bias toward the metropolitan region in the Omicron dataset. To test the potential impact of that geographic bias, we compared phylogeographic reconstructions of the most prevalent BA.1 lineage using the entire dataset and a subset where the proportion of sequences from Manaus (71%) in the entire dataset was reduced to the proportion of SARS-CoV-2 positive cases reported in Manaus over the whole Amazonas state (61%). Both analyses point to very similar patterns of viral dispersion, indicating that major findings in our study were robust to sampling bias. We thank the reviewer for bringing attention to this point. We modified the methods section to clarify the sampling procedure and included a new section in the results describing these new analyses.

MINORS COMMENTS

Fig1 A: y axis contains “?”

Reply: This mistake was rectified in the version of this figure.

Fig5: in the caption and in the figure, you used Rt instead of Re

Reply: This inconsistency was corrected in the current figure.

The use of different nomenclature (while in my opinion unavoidable in this paper) sometimes makes the reading tricky. Try to be as clear as possible and to make comparisons possible between lineages in one claim / sentence. For example, line 111 “as

Omicron (B.1.1.529/BA.1.*) and four (<1%) as B.1.621” you should specify “(Mu)” at the end of the sentence. Check all the manuscript.

Reply: Complying with the reviewer request, we check the manuscript to maintain the same nomenclature of lineage/variant classification in a single sentence.

Not sure that the colors of all figures are color blind-friendly.

Reply: A more color blind-friendly color scheme was applied to the figures in their newer versions.

Check readability of figure captions according to the size they display for big figures. For example, color of the lines of the legend of fig1 C D E and fig2 D E are difficult to see when watched at 100%. The same goes for the text of figure 4. Check all figures.

Reply: In the new versions of figures 1 and 2, all lines have an increased width to make them more readable. In the new version of figure 4, the font size was increased in most of the text.

Why isn't there Delta in Fig 2 D and E?

Reply: Because in that figure we compare the three COVID-19 exponential epidemic waves and dissemination of Delta was not associated with an exponential (or linear) growth of SARS-CoV-2 cases in Amazonas.

Line 181, BA.14.2 should probably read BA.1.14.2

Reply: Correction performed.

Specify how is CFR computed in the text and not only in fig2 caption

Reply: Complying with the reviewer request, we described in the text (last paragraph of the Epidemic dynamics of SARS-CoV-2 VOCs Delta and Omicron section) how the CFR was computed.

Make sure to make the data publicly available if not already done

Reply: All the SARS-CoV-2 genomes generated in this study are available at the EpiCoV database in GISAID (<https://www.gisaid.org/>) and can be assessed at the following link: https://epicov.org/epi3/epi_set/220913va?main=true.

REVIEWER #2

1) I have only one suggestion and several minor comments.

In the section starting on line 213 (“Identification of major Delta and Omicron transmission lineages in Amazonas”), the Authors find that Manaus acted as a major source of viral lineages to other regions within Amazonas and most large Amazonian clades likely originated in Manaus. While this makes sense epidemiologically, the majority (~80%) of Amazonas sequences within the study are from Manaus. Therefore, to enable robust conclusions about the sources and origins of viral lineages, the Authors should repeat the phylogeographic analyses with random subsamples of Manaus sequences (for example to the number of sequences from the second most common location) and ensure that the same patterns are present.

Reply: The proportion of SARS-CoV-2 genomes from the Manaus metropolitan area included in our analyses during the period of dominance of VOCs Gamma/Delta (80%) was in line with the geographic origin of most SARS-CoV-2 confirmed cases (69%) in the study period ($P > 0.05$). During the period of dominance of Omicron, by contrast, we detected a slight overrepresentation of sequences from Manaus (77%) concerning the proportion of SARS-CoV-2 cases detected in that region (61%), which is explained by the wider use of rapid SARS-CoV-2 antigen tests (not selected for sequencing) at interior municipalities compared with Manaus during the Omicron wave. We agree with the reviewer that this sampling bias might impact the phylogeographical estimates. However, in our opinion, reducing the number of sequences from Manaus to the second most common location is not the best way to correct for potential sampling bias during the Omicron wave because that strategy would introduce another sampling bias, now with respect to the overall population size of SARS-CoV-2 epidemics at different locations. The size of the epidemic is a relevant epidemiological driver of viral spread, as was demonstrated in a previous study that showed that locations with larger populations of HIV-infected individuals tend to act as major sources for onward spread (DOI: <https://doi.org/10.1128/JVI.01681-15>). Thus, to test the potential impact of geographic sampling bias in our analyses, we performed phylogeographic reconstructions of the most prevalent BA.1 lineage using the entire dataset and a random subset where the proportion of sequences from Manaus (71%) was reduced to the corresponding contribution of that location to all SARS-CoV-2 positive cases (61%) in Amazonas state. Both analyses point to very similar patterns of viral dispersion and indicate that major findings in our study were robust to sampling bias. We thank the reviewer for bringing attention to this point, and we have included a new section in methods and results describing these new analyses.

MINOR COMMENTS

Line 62 – as the Omicron section of the manuscript is focused on BA.1, it would be useful to include a description of the major Omicron lineages (BA.1, BA.2, BA.4 and BA.5) and their circulation in the introduction.

Reply: Complying with the reviewer comment, we include a brief description of the Omicron family in the introduction section.

Figure 1A – y-axis labels show “10?”, presumably should be updated

Reply: This mistake was corrected in the new version of this figure.

Figure 1A – how many days are included in each rolling date? It’s not currently clear whether bars correspond to cases or genomes (presumably cases) – is it possible to clarify in the legend?

Reply: That graphic depicts the 7-day rolling average of new confirmed SARS-CoV-2 cases. We modify the legend of figure according to the reviewer’s request.

Figure 1C – this panel would be easier to interpret if the x-axis was dates (as in 1A) rather than days

Reply: In its new version, the x-axis of the figure was changed in accordance with the reviewer's suggestion.

Figure 1F doesn’t seem to exist, was this merged into 1E?

Reply: Figure 1 was modified and that error corrected.

Figure 2D and 2E – is the Omicron colour correct in the legend?

Reply: Despite being correct, the readability of the graphs was compromised by the reduced width of their lines. This was changed in the new version of this figure.

Lines 159-162 – should state that this is the CFR for detected cases, not for overall cases

Reply: Complying with the reviewer request, we described in more detail how the CFR was computed (last paragraph of the Epidemic dynamics of SARS-CoV-2 VOCs Delta and Omicron section).

Line 178 – the Delta lineages should be B.1.617.2/AY.* rather than B.1.617/AY.*

Reply: Correction performed.

Lines 193-194 – some more discussion around the number of introductions per lineage would be useful. The number of introductions doesn’t seem to be directly comparable between lineages as lineages have different sizes and ranges. For example, BA.1 is a very large globally distributed lineage so it might be expected to be imported multiple times.

Whereas BA.1.14.2 is more localized to Brazil so international imports are expected to be rare.

Reply: The changes made in **Figures 3A, 3B, 3C, and 3D** in accordance with the reviewer's suggestion allow a better comprehension of this topic since national and foreign introductions are now discriminated by time and lineage of occurrence. The results and discussion sections were updated in the manuscript's newer versions to incorporate these results.

Figure 3A and 3B – it seems like these panels would be more informative in number of introductions rather than proportion of introductions

Reply: In compliance with the reviewer request, both figures now include the number of introductions and not the proportion of their occurrence as in their original version. Additionally, as the reviewer suggested for Figures 3C e 3D, Figures 3A and B also discriminate between national and foreign introductions.

Figure 3C and 3D – it would be useful to split this into introductions into Amazonas from elsewhere in Brazil and international introductions to show different sources for different lineages

Reply: In compliance with the reviewer request, both figures discriminate between national and foreign introductions.

Figure 4A – there seems to be a high number of virus movements from Parintins to Manaus within Delta. Can the Authors include a description of these movements and whether they are spread across transmission lineages? I.e., is that a general pattern across lineages or specific to one lineage?

Reply: Viral movements from the Parintins to the Manaus region were not a common find across all Delta lineages, representing a tiny fraction (<1%) of movements in four of the studied lineages but having the absolute majority (99.7%) of its occurrences taking place in lineage AY.99.2 that also hosted the majority of Delta migrations in the state (62%). In this lineage, the aforementioned viral migration direction (Parintins/Manaus) was the most common occurrence among all detected events in the state (32%), followed by those in the opposite direction (Manaus/Parintins, 22%). A similar pattern, however, was not observed in any of the studied VOC Omicron lineages. The one with the largest share (40%) of all Parintins/Manaus migration events detected (BA.1) had these, nonetheless, as an infrequent occurrence (1%). We modify the results and discussion sections of the manuscript to describe in more detail those differences.

Line 271/Figure 5 – lineage AY.99.2AM-V doesn't appear to be in Figure 5C, it might be useful to include as it is discussed in the text

Reply: There were indeed multiple references in the submitted manuscript to the lineage AY.99.2AM-V, both in the results and the discussion section. These were, nonetheless, erroneous, since the described behavior, that is, an early emergence, preceding all other VOC Delta clusters in Amazonas corresponds to cluster AY.99.2AM-IV, and not AY.99.2AM-V. All mentions to AY.99.2AM-V were corrected in the manuscript's new version, and we apologize for the typo error.

Lines 377-378 – the term “population immunity” seems too broad here. Are the Authors referring to immunity acquired before Delta and from vaccines targeted against the original SARS-CoV-2 lineage?

Reply: The reviewer is correct. We modify the discussion section to clarify this point.

Lines 411-414 – it seems difficult to separate the role of control measures from other factors. There are other countries that imposed restrictions in earlier waves but not with Omicron – do other countries show similar differences in curve shape between waves?

Reply: The main characteristics of the Omicron (BA.1) wave in Amazonas (higher peak of SARS-CoV-2 cases and shorter duration than previous waves) were also observed in other western countries that also imposed more stringent restrictions during pre-Omicron epidemics than during Omicron waves. We modified the discussion section to clarify this point.

Lines 476-478 – what were the parameters for excluding sequences after this reassembly?

Reply: When sequences assembled by DRAGEN COVID lineage showed more than 3% Ns, as reported by Nextclade, we attempted another contig assembly protocol using BBduk and BBMap embedded in Geneious Prime. Thus, these newly assembled contigs were carefully visually inspected, and the remaining low-quality regions in reads (if they exist) were manually trimmed. Usually, we consider having success if the number of missing information was reduced to less than <3%, as confirmed by the second round of Nextclade analysis.

Section starting on line 528 – the discrete labels used in the phylogeographic analysis and the logic behind the choice of labels should be outlined

Reply: In accordance with the reviewer's suggestion, the methodology section was updated to clarify the choice of labels made during the phylogeographic analysis.

Line 546 – seems to be a typo in “ggaaluvial” which should be “ggalluvial”

Reply: Correction performed.

REVIEWER #3

Overall, I find these findings not very surprising but interesting as they do offer some insights into the Delta and Omicron competition epidemics in a very concentrated region. However, I do think that there are several points where clarity is needed that could help strengthen the manuscript. The biggest question is regarding sampling and how was it performed and the nature of such biases. The authors do cite sampling as a limitation but they merely talk about it from a numbers perspective as oppose to the methodology of sampling: randomly chosen samples vs. dense sampling of specific outbreaks in certain facilities such as hospitals etc. This will have an effect on both the size of the transmission chains and the lag

1) How are these samples chosen for sequenced and even sampled. Is it random or is it mainly derived from hospitals. Some clarity on the nature of sampling is required and the limitations of this approach is warranted.

Reply: The Health Surveillance Foundation - Dra. Rosemary Costa Pinto (FVS-RCP/AM), through the Central Laboratory from the State of Amazonas (LACEN-AM), has been the principal responsible for the molecular diagnosis of SARS-CoV-2 since the very beginning of the COVID-19 epidemic at the Amazonas state and receives suspected samples from private and public health institutions from the entire state. Initially, all positive samples with a real-time RT-PCR cycling threshold (Ct) below 30 were qualified for sequencing. We randomly selected a subset of positive samples by epidemiological week and municipality, avoiding overrepresenting one local to the detriment of the others. Since all were eligible, there was no intentional bias considering the dense sampling of specific outbreaks in selected locations, hospitals, or other health units. We also do not consider the health status (severe, mild, asymptomatic) or the epidemiological data (sex, age) of individuals as a criterion for sampling. Thus, our sequence data represent a random sample of all SARS-CoV-2 positive cases detected in the Amazonas state in the study period with no bias other than diagnosis by RT-PCR and Ct values <30. Because rapid antigen tests for SARS-CoV-2 diagnosis were more widely used in interior municipalities than in the metropolitan region of Manaus, particularly during the Omicron wave, this introduced a slight geographical bias toward the metropolitan region in the Omicron dataset. To test the potential impact of that geographic bias, we compared phylogeographic reconstructions of the most prevalent BA.1 lineage using the entire dataset and a subset where the proportion of sequences from Manaus (71%) was reduced to the proportion of SARS-CoV-2 positive cases reported in Manaus over the whole Amazonas state (61%). Both analyses point to very similar patterns of viral dispersion, indicating that major findings in our study were robust to sampling bias. We thank the reviewer for bringing attention to this point. We modified the methods section to clarify the sampling procedure and included a new section in the results describing these new analyses.

2) The lack of temporal signal in the assembled datasets is maybe not surprising given the relatively slow evolutionary rates of SARS-CoV-2 and the short study period. However, it is not clear where the rate of $5-15 \times 10^{-4}$ substitutions/site/year comes from. Can the authors justify this with a reference or further analysis?

Reply: We select a relaxed uniform prior on substitution rates ($5-15 \times 10^{-4}$ subs/site/year) that covers previous mean estimates across different time-scales (doi:10.1093/ve/veaa061; doi:10.1093/molbev/msac009; doi:10.1093/molbev/msac013). We modified the methods section to clarify this point.

3) I applaud the authors for using the thorny version of BEAST to carry out an analysis of a bigger dataset but these xml files should be made publicly available on a github or zendo repository as I think they would be of use to many researchers and the fact that it is the trees and not sequence data that is shared within these xmls there should be no prohibitive effect of sharing from GISAID. In fact given they chose this approach they could have easily scaled their numbers up.

Reply: Complying with the reviewer's request, all XML files used in the temporal and phylogeographic analysis were made publicly available on GitHub (https://github.com/flaviviruslab/covid_am_delta_omicron). In the newer version of the manuscript, the methodology section was updated to reflect this.

4) How many inferred introduction events were singletons? Given the sequencing coverage rate this may also imply that many of these singletons may actually represent small transmission lineages assuming that the rate of undersampling is uniform across transmission lineages of all sizes. In fact, the authors should clearly define what constitutes a transmission chain as this varies depending on the phylogenetic study in question.

Reply: Most Delta ($n = 105$, 88%) and Omicron ($n = 473$, 97%) introductions in Amazonas resulted in either singleton or small clades. As the reviewer correctly points out, due to our study's low sequencing coverage, many singletons may represent minor transmission lineages, so we decided not to distinguish between singletons and minor transmission lineages. Instead, we focus only on large transmission lineages that, as described in the results section, were defined as highly supported ($aLRT > 0.80$) monophyletic clades that descend from a most recent common ancestor (MRCA) probably located (PSP [posterior state probability] > 0.90) in Amazonas and that comprises at least 1% of all Delta ($n > 10$) and Omicron ($n > 20$) Amazonian sequences. To clarify this point, we modify the results section and describe the proportion of Delta and Omicron introductions that resulted in singletons/small clades in our analyses.

5) Do the authors have any available data on vaccination rates per region and how this may correlated with their findings?

Reply: Available data on vaccination rates per region in Amazonas suggests no correlation between vaccination and the different speeds of displacement observed for VOCs Gamma and Delta (highlighted in Fig. 1), the reason why it was kept out of the submitted manuscript. The graph below illustrates the proportion of single and fully vaccinated individuals, stratified by residence location. Despite an overall comparable profile of the curves inside and outside Manaus, the proportion of single and fully vaccinated individuals was, from May 2021 onwards, consistently higher in the capital (~10%) compared to the other regions considered together. Such difference could either be negligible at the populational level or have led to a somewhat slowed-down dissemination in Manaus. The latter was not the case, as both Delta and Omicron exhibited a faster displacement of their preceding VOCs in the capital, suggesting that a causative association between vaccination level inside/outside Manaus and VOC displacement speed could not be directly established.

6) I'm assuming that there was no restriction in terms of social mixing or curfews etc. that could help explain the data during the observed study period.

Reply: During the periods of the spread of VOCs Delta and Omicron in Amazonas, the restriction in terms of social mixing was much lower than during the period of spread variants B.1 and Gamma, and this is reflected in the daily mobility data from Google COVID-19 Community Mobility Reports (Google 2022), which use aggregated and anonymized mobility data to measure the change in total visitors compared to a baseline value at different categories of places over time. By contrast, no significant differences in human mobility were observed between the periods of the spread of Delta and Omicron, indicating that differences in spread dynamics between those VOCs could not be explained in terms of differences in social mixing. We modified the discussion section and included an additional supplementary figure showing the temporal variation in the daily mobility data from 2020 to 2022.

REVIEWERS' COMMENTS

Reviewer #1 (Remarks to the Author):

Review of "Comparative epidemic expansion of SARS-CoV-2 variants Delta and Omicron in Amazonas, a Brazilian setting with high levels of hybrid immunity"

The authors have reworked the manuscript and thereby substantially increased the quality of their work.

A few typos remain and some sentences might benefit from being rewritten. Below is a non-exhaustive list, but a careful full read might highlight some extra ones.

L279 transmission and only few of them originated (from?) local transmissions lineages

L300 former had a sampling raNge

L299-301 rework sentence

Reviewer #2 (Remarks to the Author):

The Authors have done a thorough job responding to my previous comments. I have no further comments

Reviewer #3 (Remarks to the Author):

The authors have addressed all my prior concerns and revised the manuscript accordingly.

Point-by-point response to the reviewers' comments – R1

Reviewer #1 (Remarks to the Author):

Review of “Comparative epidemic expansion of SARS-CoV-2 variants Delta and Omicron in Amazonas, a Brazilian setting with high levels of hybrid immunity”

The authors have reworked the manuscript and thereby substantially increased the quality of their work.

A few typos remain and some sentences might benefit from being rewritten. Below is a non-exhaustive list, but a careful full read might highlight some extra ones.

L279 transmission and only few of them originated (from?) local transmissions lineages

L300 former had a sampling raNge

L299-301 rework sentence

Reply: We made all modifications requested by the reviewer #1.

Reviewer #2 (Remarks to the Author):

The Authors have done a thorough job responding to my previous comments. I have no further comments

Reviewer #3 (Remarks to the Author):

The authors have addressed all my prior concerns and revised the manuscript accordingly.

Reply: We appreciate all reviewers comments.